# FactorizePhys: Matrix Factorization for Multidimensional Attention in Remote Physiological Sensing

**Jitesh Joshi**[1], **Sos S. Agaian**[2], and **Youngjun Cho**[1]

[1]Department of Computer Science, University College London, UK
[2]Department of Computer Science, College of Staten Island, City University of New York, USA
{jitesh.joshi.20, youngjun.cho}@ucl.ac.uk, sos.agaian@csi.cuny.edu

## Abstract

Remote photoplethysmography (rPPG) enables non-invasive extraction of blood volume pulse signals through imaging, transforming spatial-temporal data into time series signals. Advances in end-to-end rPPG approaches have focused on this transformation where attention mechanisms are crucial for feature extraction. However, existing methods compute attention disjointly across spatial, temporal, and channel dimensions. Here, we propose the Factorized Self-Attention Module (FSAM), which jointly computes multidimensional attention from voxel embeddings using nonnegative matrix factorization. To demonstrate FSAM's effectiveness, we developed FactorizePhys, an end-to-end 3D-CNN architecture for estimating blood volume pulse signals from raw video frames. Our approach adeptly factorizes voxel embeddings to achieve comprehensive spatial, temporal, and channel attention, enhancing performance of generic signal extraction tasks. Furthermore, we deploy FSAM within an existing 2D-CNN-based rPPG architecture to illustrate its versatility. FSAM and FactorizePhys are thoroughly evaluated against state-of-the-art rPPG methods, each representing different types of architecture and attention mechanism. We perform ablation studies to investigate the architectural decisions and hyperparameters of FSAM. Experiments on four publicly available datasets and intuitive visualization of learned spatial-temporal features substantiate the effectiveness of FSAM and enhanced cross-dataset generalization in estimating rPPG signals, suggesting its broader potential as a multidimensional attention mechanism. The code is accessible at https://github.com/PhysiologicAILab/FactorizePhys.

## 1 Introduction

Attention mechanisms in computer vision are inspired by the human ability to identify salient regions in complex scenes. Such mechanisms can be interpreted as a dynamic weight adjustment process that selects useful features and disregards irrelevant ones in a multidimensional feature space. Recent surveys [20, 23] provide a comprehensive overview of attention mechanisms and distinctly categorize existing attention mechanisms. Amidst a spectrum of research from convolution block attention [66] to computationally intensive multi-head attention [58], an effective, yet computation and memory efficient, attention mechanism has remained desirable for real-world applications. Matrix decomposition [12, 19, 31], a dimensionality reduction technique, has captured the interest of researchers and has been explored in deep learning research for different objectives [57, 60, 17, 18]. This work investigates nonnegative matrix factorization (NMF), a matrix decomposition technique, for its potential to efficiently perform multidimensional attention and evaluates its effectiveness in the spatial-temporal context of estimating rPPG signal from video frames.

38th Conference on Neural Information Processing Systems (NeurIPS 2024).

Verkruysse [59]'s pioneering investigation on extracting photoplethysmography (PPG) or blood volume pulse (BVP) signals from RGB cameras in a contactless manner led to an exciting research field of imaging-based physiological sensing. There exist several potential applications and contexts of noninvasive and contactless measurement techniques, such as stress and mental workload recognition [8, 9, 7], driver drowsiness monitoring [71] and social biofeedback interaction [43]. The seminal works on unsupervised rPPG methods [59, 48, 11] either used video frames acquired under stationary conditions or performed skin segmentation [62] or region of interest (RoI) tracking as a preprocessing step. This preprocessing step can be considered as a basic form of attention mechanism that enables the unsupervised models to process only the relevant regions. Some of the supervised rPPG methods, including HR-CNN [54], RhythmNet [45], NAS-HR [40], PulseGAN [51], and Dual-GAN [41] also relied on extracting spatial-temporal features from the tracked RoIs as a preprocessing step.

As end-to-end rPPG methods, such as DeepPhys [6], and MTTS-CAN [36] among several others, take whole facial frames as input, they rely on attention mechanisms that enable models to emphasize the relevant spatial-temporal features. Estimating BVP signal from raw facial video frames in an end-to-end manner is therefore an interesting downstream task to investigate the attention mechanism in multidimensional feature space. This requires networks to learn to pick the spatial features having the desired temporal signature, while discarding the variance related to head-motion, illumination, and skin-tones, thus representing one of the challenging spatial-temporal tasks. Few other notable end-to-end rPPG methods include PhysNet [83], 3DCNN [4], SAM-rPPGNet [26], RTrPPG [3], and transformer-network-based methods such as PhysFormer [77], PhysFormer++ [76], EfficientPhys [37], JAMSNet [79], and GLISNet [80]. A recent survey article on visual contactless physiological monitoring in clinical settings [27] highlights susceptibility to disturbance, such as head movement, as one of the key challenges, among others. Some of the recent end-to-end rPPG methods [79, 80] further highlight the need for multidimensional attention, as squeezing features in selective dimensions for deriving attention reduces the feature space to a single dimension and is therefore not well suited for the task of signal extraction.

To address this, our work draws inspiration from the seminal work on NMF [31] which a recent work formulated as an approach to design the global information block, referred to as Hamburger [18]. Hamburger [18] implements NMF to derive low-rank embeddings, which serve as a global context block. Despite the low computational complexity of $O(n)$, Hamburger [18] outperformed various attention modules in the semantic segmentation [68] and image generation tasks. In addition, researchers have combined matrix factorization with deep architectures in several ways for different applications such as layer-wise learning of dictionary for classification and clustering [57], adaptive learning of dictionary for image denoising [81], multi-attention model for recommendation systems [60], and linearly scalable approach to context modeling for medical image segmentation [1] among several others. Drawing inspiration from these studies, especially those that use matrix factorization to model global context [18, 1] in vision tasks, we investigate the application of NMF as a multidimensional attention block. Although matrix factorization in deep learning has remained a topic of significant interest, it has not been investigated in the realm of rPPG, which stands to gain from joint spatial, temporal, and channel attention.

We introduce the Factorized Self-Attention Module (FSAM), which implements NMF to jointly compute spatial-temporal attention and describe an appropriate formulation for the low-rank recovery problem. To investigate the relevance and effectiveness of FSAM in computing multidimensional attention, we build a 3D-CNN architecture FactorizePhys that implements FSAM. We further adapt FSAM for EfficientPhys [37], an end-to-end rPPG architecture that builds on the Temporal Shift Module (TSM) [34], to uniquely learn spatial-temporal features using 2D-CNN layers. Evaluation of FactorizePhys and EfficientPhys [37] with FSAM, against existing SOTA rPPG methods, demonstrates the versatility of FSAM as multidimensional attention along with its effectiveness for the downstream task of estimating time series from spatial-temporal data. In summary, we make the following contributions.

- Factorized Self-Attention Module (FSAM): NMF [31]-based novel approach that jointly computes multidimensional attention within voxel embeddings.
- FactorizePhys: an end-to-end 3D-CNN architecture that integrates FSAM for robust estimation of rPPG from spatial-temporal facial video frames.
- Thorough assessment of FactorizePhys and FSAM with multiple evaluation metrics and the corresponding measure of standard errors to compare cross-dataset generalization performance with SOTA rPPG methods, using four benchmarking rPPG datasets.

## 2 Related Work

### 2.1 Attention Mechanisms in Vision

Varied forms of attention mechanisms have been successful in different visual tasks such as image classification [70, 25, 66], object detection [5, 82], semantic segmentation [78, 16, 18, 28], video understanding [63, 15, 33, 21], 3D vision [69, 24], and multimodal tasks [73, 56] among others [20, 23]. The most widely used attention mechanisms are channel attention [35, 75], spatial attention [66, 63], temporal attention [72, 74], self-attention or transformer-based approaches [58, 14], multimodal attention [56, 73], graph-based approaches [32], as well as different combinations of these types [66, 16, 53]. In addition, researchers have proposed attention mechanisms for video understanding [63, 15, 33, 21] as well as 3D vision [69, 24]. Despite notable advances in different forms of attention mechanisms, some of the existing challenges include the requirement for high computational costs, large training data, the overall efficiency of the model, and a cost-benefit analysis of performance improvement [23]. Additionally, for rPPG research, the impact of attention mechanisms on an ability of models to generalize on unseen datasets is not systematically studied, which we address in this work.

### 2.2 Attention Mechanisms in rPPG Methods

End-to-end rPPG methods can be categorized into convolution neural networks (CNN) architectures such as PhysNet [83], EfficientPhys-C [37], 3DCNN [4], SAM-rPPGNet [26], and RTrPPG [3], and transformer-network-based architectures such as PhysFormer [77], PhysFormer++ [76], and EfficientPhys-T [37]. Among end-to-end rPPG methods, DeepPhys [6] first implemented a novel convolutional attention mechanism in an architecture that comprised separate *motion* and *appearance* branches, with the latter intended to compute attention for the main *motion* branch. Inspired by the CBAM attention mechanism [66], originally validated for classification and detection tasks, ST-Attention [46] was proposed to filter salient information from spatial-temporal maps, thus improving remote HR estimation. The similar dual attention mechanism was also found to be effective in the SMP-Net framework [13], which jointly learned the features of RGB and infrared spatial-temporal maps to estimate multiple physiological signals.

Recently, EfficientPhys [37], an end-to-end network, presented an efficient single-branch approach with a gated attention mechanism. The Swin-Transformer [39] based version of EfficientPhys [37] insightfully added the TSM [34] module to the Swin transformer [39], enabling the architecture to perform efficient spatial-temporal modeling and compute attention by combining shifting window partitions spatially and shifting frames temporally. It should be noted that the convolution-based version of EfficientPhys [37], which combined the TSM [34] module and a convolutional attention mechanism [6] showed superior accuracy along with significantly low latency, making it highly suitable for deployment on mobile devices. Recently, there has been an upsurge in transformer-based rPPG architectures, some of which include PhysFormer [77], PhysFormer++ [76], TransPhys [61], and RADIANT [22].

Unlike other transformer-based architectures that rely on spatial-temporal maps as input, PhysFormer [77] and PhysFormer++ [76] are end-to-end video transformer-based architectures, which adaptively aggregate both local and global spatial-temporal features. PhysFormer++ [76] extends PhysFormer [77] by better exploiting temporal contextual and periodic rPPG clues, as it extracts and fuses attentional features from slow and fast pathways. In addition, both architectures [77, 76] are trained using label distribution learning and a curriculum learning-inspired dynamic constraint in the frequency domain, which helps to alleviate overfitting. Although unlike convolution-based EfficientPhys [37], transformer architectures require significantly higher computational resources.

Most of the light-weight convolutional attention mechanisms require attention to be separately derived in spatial, temporal, and channel dimensions, which is later merged [46, 13]. Although 3D-CNN architectures such as PhysNet [83] and iBVPNet [29] have shown promising performances, they have not explored attention mechanisms that can potentially enhance performance in unseen datasets. JAMSNet [79] and GLISNet [80] are recent 3D-CNN architectures that benefit significantly from channel-temporal joint attention (CTJA) and spatial-temporal joint attention (STJA). However, unlike CTJA and STJA [79, 80], we jointly derive attention in temporal, spatial, and channel dimensions, without squeezing any dimension of multidimensional features.

# 3   Method

## 3.1   Primer: Nonnegative Matrix Factorization

Nonnegative matrix factorization is a dimensionality reduction paradigm that decomposes $M \times N$ matrix $V = [v_1, v_2, ..., v_N] \in \mathbb{R}_{\geq 0}^{M \times N}$ into nonnegative $M \times L$ basis matrix $W = [w1, w2, ..., wL] \in \mathbb{R}_{\geq 0}^{M \times L}$ and nonnegative $L \times N$ coefficient matrix $H = [h1, h2, ..., hN] \in \mathbb{R}_{\geq 0}^{L \times N}$, as depicted in fig. 1 and expressed as:

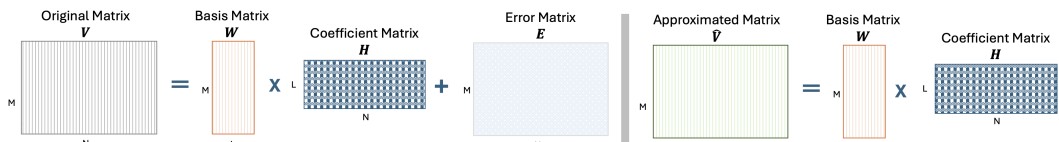

Figure 1: Formulation of Nonnegative Matrix Factorization (NMF)

$$V = WH + E = \hat{V} + E \tag{1}$$

where $\hat{V} = [\hat{v_1}, \hat{v_2}, ..., \hat{v_N}] \in \mathbb{R}_{\geq 0}^{M \times N}$ is reconstructed low-rank matrix and $E \in \mathbb{R}_{\geq 0}^{M \times N}$ is an error matrix, which is discarded. $\mathbb{R}_{\geq 0}^{M \times N}$ stands for the set of $M \times N$ element-wise nonnegative matrices. Equivalent vector formulation for this approximation can be expressed as:

$$v_j \approx \hat{v}_j = \sum_i^L w_i H_{ij} \tag{2}$$

The objective to represent high-dimensional matrix with fewer basis can be achieved only when $L$ is chosen such that $L \ll min(M, N)$, while when $L$ is larger than $M$, it results in over-complete basis. An optimization in $W$ and $H$ to achieve the optimal approximation effectively results in the discovery of inherent correlations between the basis vectors in $W$ and the corresponding coefficients in $H$ [31, 64]. The optimization objective is formulated as:

$$min_{W,H} \|V - WH\|_F^2 \ni W_{ml} \geq 0, H_{ln} \geq 0 \tag{3}$$

Further, the imposed non-negativity constraints on $W$ and $H$ enable parts-based representations, where activation of one or many of the coefficients in $H$ together with the basis vectors in $W$ can reconstruct different interpretable parts of $V$. For a detailed primer on NMF, we refer the reader to the seminal work [31] and a survey article [64] that summarizes different NMF models and algorithms.

The factorization of deeper layer embeddings can be elucidated as the squeeze of information without reducing the dimensions of the embeddings, unlike the existing attention mechanisms [25]. Therefore, exciting or multiplying with the resultant low-rank (information squeezed) embeddings can potentially serve as an attention mechanism. While factorization is formulated for two-dimensional matrix, high-dimensional embeddings can be mapped to two-dimensional matrix. In PyTorch [47], this is achieved with the 'view' operation.

## 3.2   Factorized Self-Attention Module (FSAM)

For the downstream task of estimating rPPG from video frames, spatial-temporal input data can be expressed as $\mathcal{I} \in \mathbb{R}^{T \times C \times H \times W}$, where $T, C, H, and\ W$ represents total frames (temporal dimension), channels in a frame (e.g., for RGB frames, $C = 3$), height and width of pixels in a frame, respectively. $\mathcal{I}$ is passed through a feature extractor that generates voxel embeddings $\varepsilon \in \mathbb{R}^{\tau \times \kappa \times \alpha \times \beta}$, with temporal ($\tau$), channel ($\kappa$) and spatial ($\alpha, \beta$) dimensions.

The goal is to jointly derive the attention in the multidimensional space of $\varepsilon$, without squeezing individual dimensions. For this, we deploy NMF-based matrix factorization to compute low-rank $\hat{\varepsilon}$ by reconstructing it from the factorized basis matrix $W$ and a coefficient matrix $H$. It is essential to factorize $\varepsilon$ in a way that $\hat{\varepsilon}$ approximated through the computed basis and coefficient matrices serves as an effective self-attention. Among several parameters that govern factorization, here we delve into the ones most relevant for the time series estimation task. These include: i) the transformation of the voxel embeddings that maps $\varepsilon \in \mathbb{R}^{\tau \times \kappa \times \alpha \times \beta}$ to the factorization matrix $V^{st} \in \mathbb{R}^{M \times N}$ and ii) the rank of the factorization.

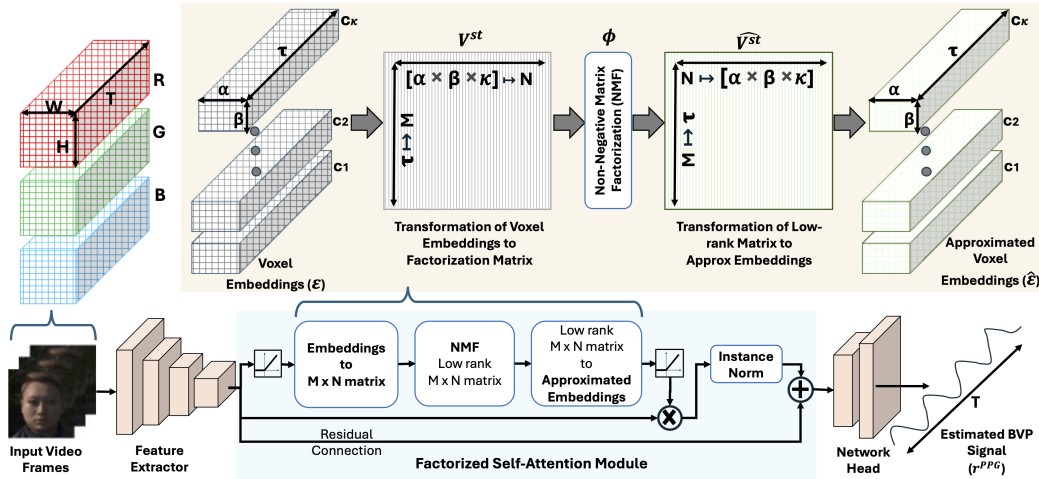

Figure 2: Factorized Self-Attention Module (FSAM) illustrated for a 3D-CNN architecture for rPPG estimation.

For a 2D-CNN architecture with $\kappa$ channels and $\alpha \times \beta$ spatial features, the transformation ($\Gamma^{\kappa\alpha\beta \mapsto MN}$) implemented in the Hamburger module [18] is expressed as:

$$V^s \in \mathbb{R}^{M \times N} = \Gamma^{\kappa\alpha\beta \mapsto MN}(\varepsilon \in \mathbb{R}^{\kappa \times \alpha \times \beta}) \ni \kappa \mapsto M, \alpha \times \beta \mapsto N \tag{4}$$

where $\kappa$ channels are mapped to $M$ and $\alpha \times \beta$ spatial features are mapped to $N$, with an underlying assumption that spatial features are inherently correlated due to learnt CNN kernels. However, for 3D-CNN architectures, as $\varepsilon \in \mathbb{R}^{\tau \times \kappa \times \alpha \times \beta}$ encodes temporal, channel, and spatial features, it is required to revisit these mappings. While it can be argued that similar to 2D-CNN architectures, as 3D-CNN architectures have 3D kernels, the spatial-temporal features are inherently correlated. However, it should be noted that the scales of spatial and temporal dimensions are very distinct, owing to which the spatial-temporal patterns to be learned may not be uniformly captured through typical convolutional kernels (e.g. $3 \times 3 \times 3$). Adjusting the spatial-temporal kernel sizes can be heuristic task, and does not guarantee the extraction of desired features, while drastically increasing the model complexity (since for time series estimation, $\tau >> \alpha, \beta$). Also, $\kappa$ channels are not inherently correlated, and therefore it is crucial to devise a multidimensional attention that jointly computes the spatial-temporal and channel attention. To address this, we first consider negative Pearson correlation, a loss function that is commonly deployed to optimize end-to-end rPPG methods, expressed as:

$$\eta_p = 1 - \frac{\sum_i^T (r_i^{ppg} - \overline{r^{ppg}})(g_i^{ppg} - \overline{g^{ppg}})}{\sqrt{\sum_i^T (r_i^{ppg} - \overline{r^{ppg}})^2} \sqrt{\sum_i^T (g_i^{ppg} - \overline{g^{ppg}})^2}} \tag{5}$$

where, $r^{ppg} \in \mathbb{R}^{1 \times T}$ is an estimated rPPG signal and $g^{ppg} \in \mathbb{R}^{1 \times T}$ corresponds to the ground-truth BVP signal. The optimization of end-to-end model to estimate a vector in temporal dimension ($r^{ppg} \in \mathbb{R}^{1 \times T}$) can be leveraged by establishing the correlation of features in spatial and channel dimensions with the features in temporal dimension. Factorization of a matrix that consists of vectors in temporal domain and spatial and channel dimension as the features of the vectors, uniquely offers an opportunity to design the requisite attention. Prior to transforming $\varepsilon$ to $V^{st} \in \mathbb{R}^{M \times N}$, it is pre-processed through a convolution layer (with $1 \times 1 \times 1$ kernels), and a ReLU activation to ensure non-negativity of the embeddings. Following this preprocessing, the temporal features of $\varepsilon$ are mapped to the vector dimension ($M$) in $V^{st}$, while spatial and channel dimensions are mapped to the feature dimension ($N$) of $V^{st}$. This transformation of $\varepsilon$ as depicted in fig. 2, can be expressed as:

$$V^{st} \in \mathbb{R}^{M \times N} = \Gamma^{\tau\kappa\alpha\beta \mapsto MN}(\xi_{pre}(\varepsilon \in \mathbb{R}^{\tau \times \kappa \times \alpha \times \beta})) \ni \tau \mapsto M, \kappa \times \alpha \times \beta \mapsto N \tag{6}$$

where, $\xi_{pre}$ represents preprocessing operation. Factorization of thus formed matrix $V^{st}$ with temporal vectors shall result in a low-rank matrix $\hat{V^{st}}$ which is approximated based on the latent structure that establishes correlation of temporal features with spatial and channel features.

$$\hat{V^{st}} = \phi(V^{st}) \tag{7}$$

where $\phi$ represents factorization operation. $\hat{V}^{st}$ is transformed back to the embedding space, resulting in an approximated voxel embeddings $\hat{\varepsilon}$ that selectively retains the spatial and channel features that contribute towards the recovery of salient temporal features in $\varepsilon$. The resultant $\hat{\varepsilon}$ can be expressed as:

$$\hat{\varepsilon} = \Gamma^{MN \mapsto \tau\kappa\alpha\beta}(\hat{V}^{st} \in \mathbb{R}^{M \times N}) \tag{8}$$

where $\Gamma^{MN \mapsto \tau\kappa\alpha\beta}$ represents matrix transformation operations. We use the one-step gradient optimization based approach [18] to factorize $V^{st}$. This approach is a linear approximation of the conventional back-propagation through time algorithm (for time $t \to \infty$) [65], as proposed with the Hamburger module [18]. Approximated low-rank matrix $\hat{V}^{st}$ is transformed back to the embedding space through $\Gamma^{MN \mapsto \tau\kappa\alpha\beta}$, resulting in $\hat{\varepsilon}$ that can potentially serve as the requisite attention. $\hat{\varepsilon}$ is post-processed with a convolution layer (with $1 \times 1 \times 1$ kernels), and a ReLU activation, followed by element-wise multiplication with $\varepsilon$. This multiplication operation serves as an excitation operation, which can be distinctly effective as $\hat{\varepsilon}$ retains the dimension of $\varepsilon$ while computing the attention. The product is instance-normalized, and added with $\varepsilon$ that serves as residual connection as depicted in fig. 2. It is to be noted that for each single forward pass through the model, approximation of $\hat{V}^{st}$ requires 4-8 steps, however, FSAM implements NMF within "no_grad" block, that does not require back-propagation through the NMF for model optimization. Representing the network head as $\omega$, $\xi_{post}$ as post-processing operation and $\mathcal{IN}$ as instance normalization, the estimated $r^{ppg}$ signal can be expressed as:

$$r^{ppg} = \omega(\varepsilon + \mathcal{IN}(\varepsilon \odot \xi_{post}(\hat{\varepsilon}))) \tag{9}$$

Next, we look at the rank of the factorization that affects the approximation of $V^{st}$. The primary consideration for the rank $L \ll min(M, N)$ as mentioned in §3.1 ensures that $\hat{V}^{st}$ is of low rank. Although the choice of $L$ is generally governed by the downstream task, it is often derived empirically. In the context of $r^{PPG}$ estimation, we revisit the formulation of factorization matrix through $\Gamma^{\tau\kappa\alpha\beta \mapsto MN}$ that maps temporal features along the $M$ dimension. As we expect only a single signal underlying source of BVP signal across all facial regions, single vector estimation $\hat{v}_0^{st}$ corresponding to rank-1 (i.e., $L = 1$) shall be sufficient to capture the spatial, temporal, and channel features that contribute to the $r^{PPG}$ estimation. Experimentation with rank-1 and higher rank factorization (appendix A.3) shows that for the higher ranks, the performance remains at par with that of the network without the FSAM, indicating that for rPPG estimation task, rank-1 factorization offers the optimal multidimensional attention, confirming our understanding.

### 3.3  Deployment of FSAM in 3D-CNN and 2D-CNN Architectures

We deploy FSAM in our proposed 3D-CNN model, FactorizePhys and integrate it in an existing 2D-CNN architecture, EfficientPhys [37] to assess its versatility.

**FactorizePhys Architecture:**  FactorizePhys, as depicted in fig. 3[A], is an end-to-end 3D-CNN architecture for estimating rPPG signal from raw video frames. Skin reflection models [62, 6] discuss the presence of several unrelated stationary and time-varying temporal components, and a relatively weaker pulsatile component of interest. To eliminate stationary components, FactorizePhys implements a `Diff` as first layer, inspired by existing rPPG architectures [6, 36, 37]. The resultant `Diff` frames are normalized with $\mathcal{IN}$, unlike existing architectures that use `BatchNorm`. The size

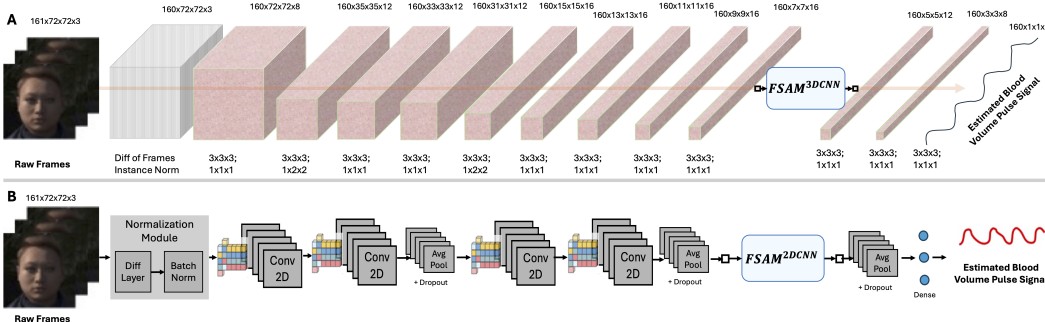

Figure 3: (A) Proposed FactorizePhys with FSAM; (B) FSAM Adapted for EfficientPhys [37]

of the kernel and the strides of each convolution layer are depicted in fig. 3[A]. For each layer, we use `TanH` activation followed by $\mathcal{IN}$. Spatial features are gradually aggregated by not padding the features, while we deploy spatial convolution strides only on the $3^{rd}$ and $6^{th}$ layers. For temporal features, *same* padding retains the input temporal dimension throughout the network, as depicted in fig. 3[A]. Downsizing of temporal features may result in high-amplitude unrelated time-varying components to outweigh the weaker rPPG related pulsatile component. To provide a clearer overview of the architecture of FactorizePhys, only the spatial and temporal dimensions of the features at multiple layers are shown in fig. 3[A], while the channel dimension is skipped. FSAM, as elaborated in §3.2, is deployed to jointly compute multidimensional attention, at the layer where the spatial dimension is reduced to $7 \times 7$, as reported as the optimal spatial dimension in a recent work [3].

**Adaptation of FSAM for 2D-CNN Architecture:**   Several SOTA rPPG methods [36, 37] leverage the TSM [34] that efficiently models spatial-temporal features using 2D-CNN architectures. The parameter called 'Frame Depth' ($\psi \ni \psi \ll \kappa$ channels) controls the number of channels that are shifted along the temporal dimension for modeling temporal features. We investigated the effectiveness of the proposed FSAM with a more recent TSM-based SOTA rPPG architecture, EfficientPhys [37], which also deploys the Self-Attention Shifted Network (SASN) as the attention module. As $\psi$ controls the amount of temporal information that is learned, we use this to formulate the mapping for the factorization matrix. Equation (6) can be adapted for TSM based architectures to appropriately transform the embeddings to factorization matrix as:

$$V^{tsm} \in \mathbb{R}^{M \times N} = \Gamma^{\kappa\alpha\beta \mapsto MN}(\varepsilon^{tsm} \in \mathbb{R}^{\kappa \times \alpha \times \beta}) \ni \psi \mapsto M, \kappa \times \alpha \times \beta \mapsto N \qquad (10)$$

Figure 3[B] shows modified EfficientPhys [37] architecture, in which we drop SASN blocks [37] and add a single FSAM. Unlike SASN [37], FSAM derives attention without squeezing any individual dimension, which in turn can strengthen the correlation between temporal, channel and spatial features. This adaption critically evaluates the proposed FSAM against its counterpart in the SOTA architecture, addressing the recommendations of a recent survey article [23] on visual attention methods.

# 4   Experiments

We perform an evaluation with carefully selected end-to-end SOTA rPPG methods that include Phys-Net [83], a 3D-CNN architecture without the attention mechanism, EfficientPhys [37], a 2D-CNN architecture with self-attention, and PhysFormer [77], a transformer-based 3D-CNN architecture with multi-head self-attention. Comparison of FactorizePhys and PhysNet [83] can indicate the importance of the attention mechanism in 3D-CNN rPPG architectures, while comparison of EfficientPhys with SASN [37] and EfficientPhys [37] with FSAM allows evaluating the effectiveness of the proposed FSAM in 2D-CNN architectures, and thus allows assessing the versatility of FSAM. Similarly, the comparison of FactorizePhys and PhysFormer [77] offers a thorough evaluation of the proposed FSAM against multi-head self-attention in 3D-CNN architectures.

Each model was trained on one of the four existing datasets that include iBVP [29], PURE [55], UBFC-rPPG [2] and SCAMPS [42] and evaluated on the other three. Appendix A.1 provides our detailed description of these datasets. Our code is based on the rPPG-Toolbox [38], with specific adaptations described in appendix A.2. We train all models uniformly with 10 epochs [79] on iBVP [29], PURE [55], and UBFC-rPPG [2] datasets, and with one epoch on SCAMPS [42] dataset. For fair evaluation, all model-specific hyperparameters were maintained as provided by the respective SOTA rPPG methods, while the training pipeline related hyperparameters, which include preprocessing steps for images and labels, batch size, number of epochs, learning rate, scheduler, and optimizer were kept consistent for training all the models.

# 5   Results and Discussion

**Ablation Study:**   First, we train FactorizePhys on UBFC-rPPG [2] dataset and test on PURE [55] and iBVP [29] datasets, to compare different transformations of $\varepsilon \in \mathbb{R}^{\tau \times \kappa \times \alpha \times \beta}$ with temporal, channel and spatial features to factorization matrix $V^{st} \in \mathbb{R}^{M \times N}$ as tabulated in table 1. Superior cross-dataset generalization can be observed when the temporal dimension, $\tau$ is mapped to $M$, as described in §3.2. We assess contribution of FSAM over base FactorizePhys model and observe consistent performance gains with FSAM, as reported in table 3 in appendix A. We then investigate

Table 1: Ablation Study for Different Mapping of Voxel Embeddings to Factorization Matrix

| Training Dataset | Testing Dataset | Mapping of Voxel Embeddings | | MAE (HR) ↓ | RMSE (HR) ↓ | MAPE (HR) ↓ | Corr (HR) ↑ | SNR (BVP) ↑ | MACC (BVP) ↑ |
|---|---|---|---|---|---|---|---|---|---|
| | | M | N | | | | | | |
| UBFC-rPPG | PURE | $\kappa$ | $\tau \times \alpha \times \beta$ | $0.77 \pm 0.42$ | $3.29 \pm 1.11$ | $1.34 \pm 0.82$ | $0.99 \pm 0.01$ | $13.84 \pm 0.82$ | $0.77 \pm 0.02$ |
| | | $\tau \times \kappa$ | $\alpha \times \beta$ | $0.71 \pm 0.39$ | $3.05 \pm 1.03$ | $1.21 \pm 0.76$ | $0.99 \pm 0.01$ | $13.60 \pm 0.81$ | $0.77 \pm 0.02$ |
| | | $\tau$ | $\alpha \times \beta \times \kappa$ | $\mathbf{0.48} \pm 0.17$ | $\mathbf{1.39} \pm 0.35$ | $\mathbf{0.72} \pm 0.28$ | $\mathbf{1.00} \pm 0.01$ | $\mathbf{14.16} \pm 0.83$ | $\mathbf{0.78} \pm 0.02$ |
| | iBVP | $\kappa$ | $\tau \times \alpha \times \beta$ | $2.05 \pm 0.40$ | $4.65 \pm 0.91$ | $2.87 \pm 0.59$ | $0.88 \pm 0.04$ | $5.99 \pm 0.58$ | $0.55 \pm 0.01$ |
| | | $\tau \times \kappa$ | $\alpha \times \beta$ | $2.17 \pm 0.46$ | $5.23 \pm 1.11$ | $3.13 \pm 0.68$ | $0.86 \pm 0.05$ | $5.83 \pm 0.57$ | $0.54 \pm 0.01$ |
| | | $\tau$ | $\alpha \times \beta \times \kappa$ | $\mathbf{1.73} \pm 0.39$ | $\mathbf{4.38} \pm 1.06$ | $\mathbf{2.40} \pm 0.57$ | $\mathbf{0.90} \pm 0.04$ | $\mathbf{6.61} \pm 0.58$ | $\mathbf{0.56} \pm 0.01$ |

residual connection in table 3, and observe it to contribute positively. We also observed that the base FactorizePhys model trained with FSAM retains the performance gains in-spite when FSAM is skipped during the inference. As this eliminates the computational overhead during inference, we report our main results of FactorizePhys trained with FSAM, by running inference without the FSAM. On contrary, in case of TSM [34] based EfficientPhys [37] model trained with FSAM, we observed performance drop when FSAM was skipped during inference, and therefore for EfficientPhys with FSAM, we do not drop FSAM during inference. Evaluation for factorization ranks and optimization steps to solve NMF shows consistent superiority of rank-1 factorization in table 4 in appendix A.

**FactorizePhys vs. State-of-the-Art:** We use heart rate (HR) [67] along with BVP metrics that include signal-to-noise ratio (SNR) and maximum amplitude of cross-correlation (MACC) [29, 10] for evaluation. SNR and MACC are direct measures to compare estimated rPPG signals with ground-truth BVP signals. The HR metrics reported are the mean absolute error (MAE), the square root of the mean square error (RMSE), the mean absolute percentage error (MAPE), and Pearson's correlation coefficient (Corr) [67] of the estimated HR. Uncertainty estimates quantifying the variability associated with signal estimation have been shown to be strongly correlated with the absolute error of the estimated HR [52]. In addition, for each metrics, we report the standard error to estimate the variability of each model. As most SOTA end-to-end models show robust within-dataset performance, we present cross-dataset performance in table 2, while reporting within-dataset performance in table 6 in appendix A.

First, we observe that for all the evaluation metrics reported, the proposed FactorizePhys with FSAM outperforms the SOTA methods on PURE [55] and iBVP [29] datasets, across all training datasets. This suggests a consistent and superior generalization achieved by the proposed method. Cross-dataset evaluation on the UBFC-rPPG [2] dataset further highlights the performance gains of the proposed FactorizePhys model when trained with the iBVP [29] and the SCAMPS [42] datasets, and at-par performance when trained with the PURE dataset. When models are trained with SCAMPS [42] (synthesized dataset), FactorizePhys uniquely outperforms the SOTA methods on all testing datasets further indicating the superior cross-dataset generalization. The performance of EfficientPhys [37] with FSAM exceeds in most cases and remains at par in the rest, compared to the EfficientPhys model with SASN [37], suggesting the versatility of FSAM as an attention module. As 3D CNN kernels in FactorizePhys can learn spatial-temporal patterns better than the TSM [34] based 2D-CNN model (EfficientPhys [37]), FactorizePhys with FSAM outperforms EfficientPhys [37] with FSAM across all datasets. Lastly, the proposed method consistently achieves superior SNR and MACC for the estimated rPPG signals, highlighting the enhanced reliability of the extracted signals.

**Computation Cost and Latency:** We compare computational complexity and latency for all the models in fig. 4[A], and provide further details in table 9 in appendix A. The cumulative MAE is computed by averaging cross-dataset performance for respective models across all combinations of training and testing datasets reported in table 2. The proposed FactorizePhys with FSAM not only shows the best performance, it has significantly less number of model parameters and performs at par in terms of latency as the 2D-CNN SOTA rPPG method, EfficientPhys [37]. Specifically, dropping FSAM during inference does not result in loss of performance for FactorizePhys, while reducing latency considerably, making it highly suitable for real-time and resource-constrained deployment. In contrast, when FSAM was dropped after training EfficientPhys [37] with FSAM, it did not retain the performance (results not shown). We interpret that while FSAM effectively influences the 3D convolutional kernels in FactorizePhys to increase the saliency of relevant spatial-temporal features, 2D convolutional kernels cannot benefit adequately due to the limited ability to model spatial-temporal

Table 2: Cross-dataset Performance Evaluation for rPPG Estimation

| Training Dataset | Model | Attention Module | MAE (HR)↓ | RMSE (HR)↓ | MAPE (HR)↓ | Corr (HR) †↑ | SNR ( dB, BVP)↑ | MACC (BVP)↑ |
|---|---|---|---|---|---|---|---|---|
| | | | Performance Evaluation on PURE Dataset | | | | | |
| iBVP | PhysNet | - | 7.78 ± 2.27 | 19.12 ± 3.93 | 8.94 ± 2.71 | 0.59 ± 0.11 | 9.90 ± 1.49 | 0.70 ± 0.03 |
| | PhysFormer | TD-MHSA* | 6.58 ± 1.98 | 16.55 ± 3.60 | 6.93 ± 1.90 | 0.76 ± 0.09 | 9.75 ± 1.96 | 0.71 ± 0.03 |
| | EfficientPhys | SASN | 0.56 ± 0.17 | 1.40 ± 0.33 | 0.87 ± 0.28 | 0.998 ± 0.01 | 11.96 ± 0.84 | 0.73 ± 0.02 |
| | EfficientPhys | FSAM (Ours) | **0.44** ± 0.14 | **1.19** ± 0.30 | **0.64** ± 0.22 | **0.999** ± 0.01 | 12.64 ± 0.78 | 0.75 ± 0.02 |
| | FactorizePhys (Ours) | FSAM (Ours) | 0.60 ± 0.21 | 1.70 ± 0.42 | 0.87 ± 0.30 | 0.997 ± 0.01 | **15.19** ± 0.91 | **0.77** ± 0.02 |
| SCAMPS | PhysNet | - | 26.74 ± 3.17 | 36.19 ± 5.18 | 46.73 ± 5.66 | 0.45 ± 0.12 | -2.21 ± 0.66 | 0.31 ± 0.02 |
| | PhysFormer | TD-MHSA* | 16.64 ± 2.95 | 28.13 ± 5.00 | 30.58 ± 5.72 | 0.51 ± 0.11 | 0.84 ± 1.00 | 0.42 ± 0.02 |
| | EfficientPhys | SASN | 6.21 ± 2.26 | 18.45 ± 4.54 | 12.16 ± 4.57 | 0.74 ± 0.09 | 4.39 ± 0.78 | 0.51 ± 0.02 |
| | EfficientPhys | FSAM (Ours) | 8.03 ± 2.25 | 19.09 ± 4.27 | 15.12 ± 4.44 | 0.73 ± 0.09 | 3.81 ± 0.79 | 0.48 ± 0.02 |
| | FactorizePhys (Ours) | FSAM (Ours) | **5.43** ± 1.93 | **15.80** ± 3.58 | **11.10** ± 4.05 | **0.80** ± 0.08 | **11.40** ± 0.76 | **0.67** ± 0.02 |
| UBFC-rPPG | PhysNet | - | 10.38 ± 2.40 | 21.14 ± 3.90 | 20.91 ± 4.97 | 0.66 ± 0.10 | 11.01 ± 0.97 | 0.72 ± 0.02 |
| | PhysFormer | TD-MHSA* | 8.90 ± 2.15 | 18.77 ± 3.67 | 17.68 ± 4.52 | 0.71 ± 0.09 | 8.73 ± 1.02 | 0.66 ± 0.02 |
| | EfficientPhys | SASN | 4.71 ± 1.79 | 14.52 ± 3.65 | 7.63 ± 2.97 | 0.80 ± 0.08 | 8.77 ± 1.00 | 0.66 ± 0.02 |
| | EfficientPhys | FSAM (Ours) | 3.69 ± 1.66 | 13.27 ± 3.55 | 5.85 ± 2.63 | 0.83 ± 0.07 | 9.65 ± 0.90 | 0.68 ± 0.02 |
| | FactorizePhys (Ours) | FSAM (Ours) | **0.48** ± 0.17 | **1.39** ± 0.35 | **0.72** ± 0.28 | **0.998** ± 0.01 | **14.16** ± 0.83 | **0.78** ± 0.02 |
| | | | Performance Evaluation on UBFC-rPPG Dataset | | | | | |
| iBVP | PhysNet | - | 3.09 ± 1.79 | 10.72 ± 4.24 | 2.83 ± 1.44 | 0.81 ± 0.10 | 7.13 ± 1.53 | 0.81 ± 0.02 |
| | PhysFormer | TD-MHSA* | 9.88 ± 2.95 | 19.59 ± 5.35 | 8.72 ± 2.42 | 0.44 ± 0.16 | 2.80 ± 2.21 | 0.70 ± 0.03 |
| | EfficientPhys | SASN | 1.14 ± 0.45 | 2.85 ± 0.88 | 1.42 ± 0.58 | 0.987 ± 0.03 | 8.71 ± 1.23 | 0.84 ± 0.01 |
| | EfficientPhys | FSAM (Ours) | 1.17 ± 0.46 | 2.87 ± 0.88 | 1.31 ± 0.53 | 0.987 ± 0.03 | 8.54 ± 1.26 | 0.85 ± 0.01 |
| | FactorizePhys (Ours) | FSAM (Ours) | **1.04** ± 0.38 | **2.40** ± 0.69 | **1.23** ± 0.48 | **0.990** ± 0.03 | **8.84** ± 1.31 | **0.86** ± 0.01 |
| PURE | PhysNet | - | 1.23 ± 0.41 | 2.65 ± 0.70 | 1.42 ± 0.50 | 0.988 ± 0.03 | 8.34 ± 1.22 | 0.85 ± 0.01 |
| | PhysFormer | TD-MHSA* | **1.01** ± 0.38 | **2.40** ± 0.69 | **1.21** ± 0.48 | **0.990** ± 0.03 | 8.42 ± 1.24 | 0.85 ± 0.01 |
| | EfficientPhys | SASN | 1.41 ± 0.49 | 3.16 ± 0.93 | 1.68 ± 0.64 | 0.982 ± 0.03 | 6.87 ± 1.15 | 0.79 ± 0.02 |
| | EfficientPhys | FSAM (Ours) | 1.20 ± 0.46 | 2.92 ± 0.92 | 1.50 ± 0.63 | 0.986 ± 0.03 | 7.37 ± 1.20 | 0.79 ± 0.01 |
| | FactorizePhys (Ours) | FSAM (Ours) | 1.04 ± 0.38 | 2.44 ± 0.69 | 1.23 ± 0.48 | 0.989 ± 0.03 | **8.88** ± 1.30 | **0.87** ± 0.01 |
| SCAMPS | PhysNet | - | 11.24 ± 2.63 | 18.81 ± 4.71 | 13.55 ± 3.81 | 0.38 ± 0.17 | -0.09 ± 1.02 | 0.48 ± 0.03 |
| | PhysFormer | TD-MHSA* | 8.42 ± 2.72 | 17.73 ± 5.09 | 11.27 ± 4.24 | 0.49 ± 0.16 | 2.29 ± 1.33 | 0.61 ± 0.03 |
| | EfficientPhys | SASN | 2.18 ± 0.75 | 4.82 ± 1.43 | 2.35 ± 0.76 | 0.96 ± 0.05 | 4.40 ± 1.03 | 0.67 ± 0.01 |
| | EfficientPhys | FSAM (Ours) | 2.69 ± 0.77 | 5.20 ± 1.39 | 3.16 ± 0.95 | 0.95 ± 0.06 | 3.74 ± 1.16 | 0.63 ± 0.02 |
| | FactorizePhys (Ours) | FSAM (Ours) | **1.17** ± 0.40 | **2.56** ± 0.70 | **1.35** ± 0.49 | **0.989** ± 0.03 | **8.41** ± 1.19 | **0.82** ± 0.01 |
| | | | Performance Evaluation on iBVP Dataset | | | | | |
| PURE | PhysNet | - | **1.63** ± 0.33 | 3.77 ± 0.73 | **2.17** ± 0.42 | 0.92 ± 0.04 | 6.08 ± 0.62 | 0.55 ± 0.01 |
| | PhysFormer | TD-MHSA* | 2.50 ± 0.64 | 7.09 ± 1.50 | 3.39 ± 0.82 | 0.79 ± 0.06 | 5.21 ± 0.60 | 0.52 ± 0.01 |
| | EfficientPhys | SASN | 3.80 ± 1.38 | 14.82 ± 3.74 | 5.15 ± 1.87 | 0.56 ± 0.08 | 2.93 ± 0.48 | 0.45 ± 0.01 |
| | EfficientPhys | FSAM (Ours) | 2.10 ± 0.33 | 4.00 ± 0.64 | 2.94 ± 0.49 | 0.91 ± 0.04 | 4.19 ± 0.54 | 0.49 ± 0.01 |
| | FactorizePhys (Ours) | FSAM (Ours) | 1.66 ± 0.30 | **3.55** ± 0.65 | 2.31 ± 0.46 | **0.93** ± 0.04 | **6.78** ± 0.57 | **0.58** ± 0.01 |
| SCAMPS | PhysNet | - | 31.85 ± 1.89 | 37.40 ± 3.38 | 45.62 ± 2.96 | -0.10 ± 0.10 | -6.11 ± 0.22 | 0.16 ± 0.00 |
| | PhysFormer | TD-MHSA* | 41.73 ± 1.31 | 43.89 ± 3.11 | 58.56 ± 2.36 | 0.15 ± 0.10 | -9.13 ± 0.53 | 0.14 ± 0.00 |
| | EfficientPhys | SASN | 26.19 ± 3.47 | 44.55 ± 6.18 | 38.11 ± 5.21 | -0.12 ± 0.10 | -2.36 ± 0.38 | 0.30 ± 0.01 |
| | EfficientPhys | FSAM (Ours) | 13.40 ± 1.69 | 22.10 ± 3.00 | 19.93 ± 2.67 | 0.05 ± 0.10 | -3.46 ± 0.26 | 0.24 ± 0.01 |
| | FactorizePhys (Ours) | FSAM (Ours) | **2.71** ± 0.54 | **6.22** ± 1.38 | **3.87** ± 0.80 | **0.81** ± 0.06 | **2.36** ± 0.47 | **0.43** ± 0.01 |
| UBFC-rPPG | PhysNet | - | 3.18 ± 0.67 | 7.65 ± 1.46 | 4.84 ± 1.14 | 0.70 ± 0.07 | 5.54 ± 0.61 | **0.56 ± 0.01** |
| | PhysFormer | TD-MHSA* | 7.86 ± 1.46 | 17.13 ± 2.69 | 11.44 ± 2.25 | 0.38 ± 0.09 | 1.71 ± 0.56 | 0.43 ± 0.01 |
| | EfficientPhys | SASN | 2.74 ± 0.63 | 7.07 ± 1.81 | 4.02 ± 1.08 | 0.74 ± 0.07 | 4.03 ± 0.55 | 0.49 ± 0.01 |
| | EfficientPhys | FSAM (Ours) | 2.56 ± 0.54 | 6.13 ± 1.32 | 3.71 ± 0.92 | 0.79 ± 0.06 | 4.65 ± 0.56 | 0.50 ± 0.01 |
| | FactorizePhys (Ours) | FSAM (Ours) | **1.74** ± 0.39 | **4.39** ± 1.06 | **2.42** ± 0.57 | **0.90** ± 0.04 | **6.59** ± 0.57 | **0.56** ± 0.01 |

TD-MHSA*: Temporal Difference Multi-Head Self-Attention [77]; SASN: Self-Attention Shifted Network [37]; FSAM: Proposed Factorized Self-Attention Module.

† All metrics are shown with two decimal places, except those correlation measures that range between 0.99 and 1.0

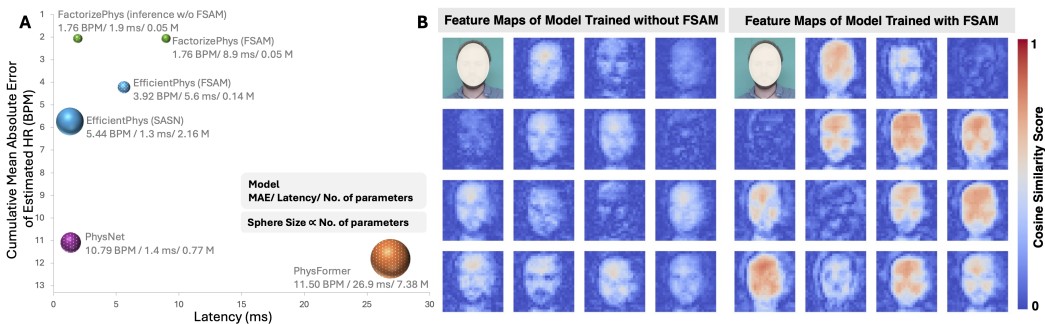

Figure 4: (A) Cumulative cross-dataset performance (MAE) v/s latency† plot. The size of the sphere corresponds to the number of model parameters; (B) Visualization of learned spatial-temporal features from the base 3D-CNN model trained without and with FSAM; † System specs: Ubuntu 22.04 OS, NVIDIA GeForce RTX 3070 Laptop GPU, Intel® Core™ i7-10870H CPU @ 2.20GHz, 16 GB RAM.

features. It should also be noted that the higher latency of FactorizePhys compared to EfficientPhys [37], although it has fewer model parameters, can be attributed to the difference in floating-point operations (FLOPS) between the 3D-CNN and 2D-CNN architectures.

**Visualization of Learned Attention:** We compute absolute cosine similarity between the temporal dimension of 4D embeddings (with temporal, spatial, and channel dimensions) and the ground-truth signal to visualize the learned attention for FactorizePhys trained without and with FSAM in fig. 4[B], where each tile represents a channel of the embedding layer. A higher cosine similarity score between the temporal dimension of the embeddings and the ground-truth PPG signal, which is observed for FactorizePhys trained with FSAM, indicates a higher saliency of temporal features. The spatial spread of high cosine similarity scores in different channels for FactorizePhys trained with FSAM, highlights selectivity of the learned attention, providing clearer evidence that the FactorizePhys model trained with FSAM can effectively pick the spatial features having the strong presence of the rPPG signal (i.e., facial regions with visible skin surface). Figure 4[B] not only suggests the effectiveness of the joint computation of multidimensional attention, but also offers more intuitive visualization of learned spatial-temporal features than existing visualization approaches [79, 37].

## 6 Conclusion

We present FactorizePhys, a 3D-CNN model utilizing the Factorized Self-Attention Module, FSAM, to concurrently extract multidimensional (spatial, temporal, and channel) attention for the downstream task of rPPG estimation from video frames. The assessment performed utilizing various rPPG datasets demonstrates that our proposed method possesses superior generalization capabilities across different datasets, compared to current state-of-the-art methods. Moreover, when adjusted to the 2D-CNN architecture, FSAM achieves performance on par with the established SASN [37] attention, underscoring its adaptability across diverse network architectures.

**Broader Impacts and Limitations:** The superior performance of FactorizePhys equipped with FSAM to estimate rPPG indicates its potential utility in various healthcare applications that require the estimation of physiological signals through noncontact imaging. Although FSAM has shown efficacy as a multidimensional attention mechanism specifically for the extraction of rPPG signals, more research is needed to determine the efficacy of the proposed method in extracting heart rate variability metrics as well as other physiological signals. Despite the state-of-the-art performance of the proposed rPPG method, signal peaks can still be susceptible to challenging real-world scenarios, such as active head movements, occlusions, and dynamic changes in ambient lighting conditions, an issue that is qualitatively illustrated in the waveforms depicted in appendix A.11. Moreover, it is imperative to conduct additional research to evaluate the effectiveness of FSAM across other spatial-temporal domains, including video understanding, video object tracking, and video segmentation, along with several other downstream tasks that depend on multi-dimensional input data. In the context of signal estimation tasks, the utilization of NMF variants that integrate temporal or frequency constraints on time series vectors may offer enhanced attention capabilities. These constraints are congruent with the characteristics of the ground truth and present avenues for future investigation.

## Acknowledgments and Disclosure of Funding

The author JJ was fully supported by the UCL CS PhD Studentship (GDI - Physiological Computing and Artificial Intelligence) which Prof. Cho has secured.

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

# A Appendix / Supplemental Material

## A.1 Datasets

All datasets provide video recordings with a resolution of $640 \times 480$, and frame rate of 30 FPS. Below we provide data-specific details.

**iBVP [29]:** The iBVP dataset consists of 124 synchronized RGB and thermal infrared videos from 31 subjects, acquired under controlled conditions. Each video is 3 minutes in duration, and the ground truth BVP signals were acquired from the ear using PhysioKit [30]. Data were acquired under 4 different conditions that include controlled breathing, math tasks, and head movements. BVP signals are marked with the signal quality, enabling the use of the video frames only where the quality of ground-truth BVP signal is high. In this work, we use only RGB frames to train the models.

**PURE [55]:** This data set comprises video recordings from 10 subjects, with the ground-truth BVP and SpO2 signals acquired from the subject's finger. For each participant, six recordings are acquired under varied motion conditions, offering a range of data reflecting different physical states.

**UBFC-rPPG [2]:** This data set contains video recordings of 43 subjects acquired under indoor conditions with a combination of natural sunlight and artificial illumination.

**SCAMPS [42]:** This dataset comprises 2800 videos of synthetic avatars that were generated through high-fidelity, quasi-photorealistic renderings. Although the videos introduce various conditions such as head motions, facial expressions, and changes in ambient illumination, they are often used as a training set rather than a validation or test set.

## A.2 Implementation Overview

The preprocessing steps for video frames include face detection using the YOLO5Face [49] face detector at an interval of 30 frames and using the detected facial bounding box to crop 30 subsequent frames, prior to performing the next face detection. The cropped facial frames are resized to a resolution of $72 \times 72$, which has been shown to be sufficient to estimate the rPPG. Additionally, to ensure uniform input data for all models, we add `Diff` layer to the PhysNet [83] and PhysFormer [77] architectures, as implemented by EfficientPhys [37] and the proposed FactorizePhys models, and train all the models from scratch using uniformly preprocessed video frames.

The number of frames in a video chunk is maintained as 161, which after the `Diff` layer becomes 160, making the spatial-temporal input data size $160 \times 72 \times 72$. Ground-truth BVP signals are also uniformly standardized for training all models. This is different from some of the recent work [37] that applies `Diff` in addition to standardization. We empirically found that all models perform significantly better when trained with the standardized BVP signals, although when the `Diff` is applied to the video frames.

All models were trained with 10 epochs on, following a recent work [79], as a higher number of epochs, e.g. 30 epochs as used in rPPG-Toolbox [38] resulted in poor generalization for all models. However, we used only one epoch for all models to train on the SCAMPS [42] dataset, since this dataset is a synthesized dataset with generated BVP signals that are easier for models to learn, unlike real-world datasets. Training beyond one epoch resulted in poorer cross-dataset performance for all the models. The batch size of 4 was used consistently throughout the training and the maximum learning rate was set to $1 \times 10^{-3}$ with 1 cycle learning rate scheduler [50] for all CNN models.

In addition, CNN models were optimized using negative Pearson correlation as a loss function. The learning rate for PhysFormer [77] was set to $1 \times 10^{-4}$ and it was optimized using a dynamic loss composed of several hyperparameters, a negative Pearson loss, a frequency cross-entropy loss and a label distribution loss as used by the authors and implemented in the rPPG-Toolbox [38]. Before computing HR for performance evaluation, both ground truth and estimated BVP signals were filtered using a bandpass filter (low cutoff = 0.60 Hz, high cutoff = 3.30 Hz) to accommodate HR ranges of 36 to 198 BPM. HR was then computed using the FFT-peaks-based approach as implemented in rPPG-Toolbox [38].

### A.3 Ablation Studies for FactorizePhys

We conduct ablation studies to evaluate optimal architectural choices and hyperparameters for the proposed FactorizePhys and FSAM. In table 3, we compare base FactorizePhys without FSAM and with FSAM and observe consistent performance gains with FSAM. Evaluation with and without residual connection indicates performance gains when residual connection around FSAM is implemented.

Table 3: Ablation study to assess residual connection to FSAM Module, and to compare the models trained with FSAM, for their inferences without FSAM

| Training Dataset | Testing Dataset | Training | Inference | MAE (HR) ↓ | RMSE (HR) ↓ | MAPE (HR) ↓ | Corr (HR) ↑ | SNR (BVP) ↑ | MACC (BVP) ↑ |
|---|---|---|---|---|---|---|---|---|---|
| UBFC-rPPG | PURE | Base | Base | 1.37 ± 1.02 | 7.97 ± 2.84 | 2.55 ± 2.07 | 0.94 ± 0.04 | 13.74 ± 0.81 | 0.77 ± 0.02 |
| | | Base + FSAM | Base + FSAM | 0.71 ± 0.39 | 3.05 ± 1.02 | 1.20 ± 0.76 | 0.99 ± 0.02 | 13.78 ± 0.81 | 0.77 ± 0.02 |
| | | | Base | 0.71 ± 0.39 | 3.05 ± 1.02 | 1.20 ± 0.76 | 0.99 ± 0.02 | 13.78 ± 0.81 | 0.77 ± 0.02 |
| | | Base + FSAM + Res | Base + FSAM + Res | **0.48 ± 0.17** | **1.39 ± 0.35** | **0.72 ± 0.28** | **1.00 ± 0.01** | **14.16 ± 0.83** | **0.78 ± 0.02** |
| | | | Base | **0.48 ± 0.17** | **1.39 ± 0.35** | **0.72 ± 0.28** | **1.00 ± 0.01** | **14.16 ± 0.83** | **0.78 ± 0.02** |
| | iBVP | Base | Base | 1.99 ± 0.42 | 4.82 ± 1.03 | 2.89 ± 0.69 | 0.87 ± 0.05 | 5.88 ± 0.57 | 0.54 ± 0.01 |
| | | Base + FSAM | Base + FSAM | 1.90 ± 0.34 | 3.99 ± 0.76 | 2.66 ± 0.50 | **0.91 ± 0.04** | 5.82 ± 0.57 | 0.54 ± 0.01 |
| | | | Base | 1.85 ± 0.33 | **3.89 ± 0.75** | 2.59 ± 0.49 | **0.91 ± 0.04** | 5.80 ± 0.57 | 0.54 ± 0.01 |
| | | Base + FSAM + Res | Base + FSAM + Res | **1.73 ± 0.39** | 4.38 ± 1.06 | **2.40 ± 0.57** | 0.90 ± 0.04 | **6.61 ± 0.58** | **0.56 ± 0.01** |
| | | | Base | 1.74 ± 0.39 | 4.39 ± 1.06 | 2.42 ± 0.57 | 0.90 ± 0.04 | 6.59 ± 0.57 | **0.56 ± 0.01** |

Retention of performance gains despite FSAM being skipped during inference, for FactorizePhys trained with FSAM offers insight into the mechanics of how FSAM functions. This can be interpreted as follows: Optimization of a network having FSAM implemented as an attention mechanism influences the network to increase the saliency of the most relevant features, so that a factorized approximation of embeddings retains these features, while discarding the less important features. Due to the increased saliency of relevant features and the presence of residual connection, FSAM can be skipped during inference, significantly reducing computational overhead.

Table 4: Performance Evaluation of Models on PURE Dataset [55], Trained with UBFC-rPPG Dataset [2], using Different Ranks and Optimization Steps for Factorization

| Optimization Steps for Matrix Factorization | Rank | MAE (HR) ↓ | | RMSE (HR) ↓ | | MAPE (HR) ↓ | | Corr (HR) ↑ | | SNR ( dB, BVP) ↑ | | MACC (BVP) ↑ | |
|---|---|---|---|---|---|---|---|---|---|---|---|---|---|
| | | Mean | SE | Mean | SE | Mean | SE | Mean | SE | Mean | SE | Mean | SE |
| Base | | 1.37 | 1.02 | 7.97 | 2.84 | 2.55 | 2.07 | 0.94 | 0.04 | 13.74 | 0.81 | 0.77 | 0.02 |
| 4 | 1 | 0.48 | 0.17 | 1.39 | 0.35 | 0.72 | 0.28 | 1.00 | 0.01 | 14.16 | 0.83 | 0.78 | 0.02 |
| | 2 | 1.40 | 1.02 | 7.98 | 2.84 | 2.59 | 2.07 | 0.94 | 0.04 | 13.71 | 0.81 | 0.77 | 0.02 |
| | 4 | 2.25 | 1.30 | 10.23 | 3.09 | 4.36 | 2.66 | 0.91 | 0.05 | 13.50 | 0.82 | 0.77 | 0.02 |
| | 8 | 1.44 | 1.02 | 7.98 | 2.84 | 2.64 | 2.07 | 0.94 | 0.04 | 13.70 | 0.83 | 0.77 | 0.02 |
| | 16 | 2.20 | 1.30 | 10.22 | 3.09 | 4.26 | 2.66 | 0.91 | 0.05 | 13.55 | 0.82 | 0.77 | 0.02 |
| 6 | 1 | 0.80 | 0.39 | 3.11 | 1.03 | 1.33 | 0.77 | 0.99 | 0.02 | 13.60 | 0.81 | 0.77 | 0.02 |
| | 2 | 1.31 | 0.84 | 6.55 | 2.30 | 2.45 | 1.72 | 0.96 | 0.04 | 13.42 | 0.81 | 0.76 | 0.02 |
| | 4 | 1.53 | 0.90 | 7.10 | 2.32 | 2.91 | 1.86 | 0.96 | 0.04 | 13.54 | 0.82 | 0.77 | 0.02 |
| | 8 | 2.22 | 1.30 | 10.23 | 3.09 | 4.29 | 2.66 | 0.91 | 0.05 | 13.75 | 0.82 | 0.77 | 0.02 |
| | 16 | 1.43 | 1.02 | 7.98 | 2.84 | 2.65 | 2.07 | 0.94 | 0.04 | 13.62 | 0.81 | 0.77 | 0.02 |
| 8 | 1 | 0.73 | 0.39 | 3.06 | 1.02 | 1.24 | 0.77 | 0.99 | 0.02 | 13.67 | 0.81 | 0.77 | 0.02 |
| | 2 | 1.44 | 1.02 | 7.98 | 2.84 | 2.64 | 2.07 | 0.94 | 0.04 | 13.35 | 0.82 | 0.77 | 0.02 |
| | 4 | 0.78 | 0.39 | 3.10 | 1.03 | 1.30 | 0.77 | 0.99 | 0.02 | 13.77 | 0.80 | 0.77 | 0.02 |
| | 8 | 0.73 | 0.39 | 3.06 | 1.02 | 1.24 | 0.77 | 0.99 | 0.02 | 13.50 | 0.82 | 0.77 | 0.02 |
| | 16 | 0.73 | 0.39 | 3.06 | 1.02 | 1.24 | 0.77 | 0.99 | 0.02 | 13.55 | 0.83 | 0.77 | 0.02 |

In table 4, we present results to compare the performance obtained for different ranks $L$, as well as the optimization steps used to solve factorization. For all experiments, FactorizePhys is trained with the UBFC-rPPG dataset [2] and the performance is presented for the PURE dataset [55]. We can observe that the best performance was achieved for rank $L = 1$ for the different steps used to solve the factorization. For higher ranks, performance remains on par with that of the network without the FSAM, indicating that for the rPPG estimation task, the rank-1 factorization offers the optimal spatial-temporal attention. These results align with the expected single source of the underlying BVP signals in different facial regions.

## A.4 Statistical Significance of the Main Results

We performed repeated experiments with 10 different random seed values between 1 and 1000 to compare the proposed FactorizePhys trained with FSAM with the best performing SOTA rPPG method. For the cross-dataset generalization results reported in table 2, EfficientPhys with SASN [37] was found to perform the best among the existing SOTA methods.

Table 5: Performance Evaluation of Models on PURE Dataset, Trained with UBFC-rPPG Dataset, using Different Random Seed Values

| Model | Random Seed Value | MAE (HR)↓ | | RMSE (HR)↓ | | MAPE (HR)↓ | | Corr (HR)↑ | | SNR ( dB, BVP)↑ | | MACC (BVP)↑ | |
|---|---|---|---|---|---|---|---|---|---|---|---|---|---|
| | | Mean | SE | Mean | SE | Mean | SE | Mean | SE | Mean | SE | Mean | SE |
| | 10 | 3.75 | 1.62 | 12.97 | 3.53 | 5.69 | 2.52 | 0.84 | 0.07 | 8.60 | 0.99 | 0.65 | 0.02 |
| | 38 | 4.46 | 1.74 | 14.09 | 3.59 | 7.18 | 2.88 | 0.81 | 0.08 | 8.69 | 1.01 | 0.66 | 0.02 |
| | 55 | 4.67 | 1.79 | 14.55 | 3.65 | 7.50 | 2.97 | 0.80 | 0.08 | 8.69 | 1.01 | 0.66 | 0.02 |
| | 100 | 4.71 | 1.79 | 14.52 | 3.65 | 7.63 | 2.97 | 0.80 | 0.08 | 8.77 | 1.00 | 0.66 | 0.02 |
| EfficientPhys with | 128 | 4.74 | 1.79 | 14.52 | 3.65 | 7.68 | 2.97 | 0.80 | 0.08 | 8.84 | 0.99 | 0.66 | 0.02 |
| SASN Attention Module | 138 | 4.36 | 1.79 | 14.41 | 3.65 | 7.01 | 2.97 | 0.80 | 0.08 | 8.81 | 0.99 | 0.66 | 0.02 |
| | 212 | 4.52 | 1.78 | 14.42 | 3.65 | 7.37 | 2.96 | 0.80 | 0.08 | 8.64 | 0.99 | 0.66 | 0.02 |
| | 308 | 4.70 | 1.79 | 14.52 | 3.65 | 7.61 | 2.97 | 0.80 | 0.08 | 8.84 | 1.03 | 0.66 | 0.02 |
| | 319 | 4.70 | 1.79 | 14.55 | 3.65 | 7.55 | 2.97 | 0.80 | 0.08 | 8.96 | 1.00 | 0.66 | 0.02 |
| | 900 | 4.63 | 1.79 | 14.51 | 3.65 | 7.48 | 2.97 | 0.80 | 0.08 | 8.65 | 0.99 | 0.66 | 0.02 |
| | **Average** | 4.52 | 1.77 | 14.31 | 3.63 | 7.27 | 2.92 | 0.81 | 0.08 | 8.75 | 1.00 | 0.66 | 0.02 |
| | 10 | 1.38 | 0.98 | 7.64 | 2.71 | 2.52 | 1.98 | 0.95 | 0.04 | 13.40 | 0.82 | 0.75 | 0.02 |
| | 38 | 4.31 | 1.86 | 14.93 | 3.79 | 7.11 | 3.18 | 0.79 | 0.08 | 12.52 | 0.84 | 0.75 | 0.02 |
| | 55 | 2.17 | 1.30 | 10.22 | 3.09 | 4.22 | 2.66 | 0.91 | 0.05 | 13.71 | 0.83 | 0.77 | 0.02 |
| | 100 | 0.48 | 0.17 | 1.39 | 0.35 | 0.72 | 0.28 | 1.00 | 0.01 | 14.16 | 0.83 | 0.78 | 0.02 |
| Proposed FactorizePhys | 128 | 0.78 | 0.39 | 3.08 | 1.03 | 1.31 | 0.77 | 0.99 | 0.02 | 13.23 | 0.81 | 0.76 | 0.02 |
| with FSAM Attention Module | 138 | 0.52 | 0.19 | 1.56 | 0.40 | 0.72 | 0.27 | 1.00 | 0.01 | 13.03 | 0.80 | 0.76 | 0.02 |
| | 212 | 2.15 | 1.22 | 9.63 | 2.88 | 4.19 | 2.50 | 0.92 | 0.05 | 13.58 | 0.81 | 0.77 | 0.02 |
| | 308 | 1.50 | 0.98 | 7.70 | 2.71 | 2.79 | 1.99 | 0.95 | 0.04 | 13.39 | 0.82 | 0.77 | 0.02 |
| | 319 | 1.38 | 0.84 | 6.61 | 2.30 | 2.60 | 1.73 | 0.96 | 0.04 | 13.54 | 0.81 | 0.77 | 0.02 |
| | 900 | 3.34 | 1.70 | 13.46 | 3.69 | 5.21 | 2.78 | 0.83 | 0.07 | 12.76 | 0.83 | 0.76 | 0.02 |
| | **Average** | 1.80 | 0.96 | 7.62 | 2.30 | 3.14 | 1.81 | 0.93 | 0.04 | 13.33 | 0.82 | 0.76 | 0.02 |
| **Paired T Test** | | 0.0001 | | 0.0014 | | 0.0001 | | 0.0004 | | 0.0000 | | 0.0000 | |

For each random seed value, we trained the proposed FactorizePhys with FSAM and EfficientPhys with SASN [37] on the UBFC-rPPG [2] dataset and evaluated them on the PURE dataset [55]. Paired T tests for each reported evaluation metrics suggest that the performance gains achieved with the proposed method are statistically significant compared against the best performing SOTA rPPG method, highlighting its effectiveness and thereby highlighting contributions of this work in the research field of end-to-end rPPG estimation from video frames.

## A.5 Within Dataset Performance

In this work, we primarily focus on comparing rPPG methods for their cross-dataset generalization, which offers more critical evaluation and reliable estimates of how models perform on unseen or

out-of-distribution data. Within-dataset performance signifies an representation ability of model to fit the data, derived from the same distribution, serving as an essential criteria. Therefore, for completeness, in table 6, we report within-dataset evaluation on iBVP [29], [55], and UBFC-rPPG [2] datasets, where we observe at-par performance of FactorizePhys as compared with the SOTA rPPG methods.

Table 6: Within Dataset Performance Evaluation

| Model | Attention Module | MAE (HR) ↓ | RMSE (HR) ↓ | MAPE (HR) ↓ | Corr (HR) ↑ | SNR ( dB, BVP) ↑ | MACC (BVP) ↑ |
|---|---|---|---|---|---|---|---|
| Performance Evaluation on iBVP Dataset, Subject-wise Split: Training (0.0 - 0.7), Test (0.7 - 1.0) | | | | | | | |
| PhysNet | - | 1.18 ± 0.29 | **2.10 ± 0.51** | 1.64 ± 0.42 | **0.98 ± 0.03** | 10.63 ± 1.05 | **0.68 ± 0.02** |
| PhysFormer | TD-MHSA* | 1.96 ± 0.63 | 4.22 ± 1.47 | 2.49 ± 0.72 | 0.91 ± 0.07 | **10.72 ± 1.04** | 0.66 ± 0.03 |
| EfficientPhys | SASN | 2.74 ± 0.96 | 6.28 ± 2.14 | 3.56 ± 1.13 | 0.81 ± 0.10 | 7.01 ± 1.03 | 0.58 ± 0.03 |
| EfficientPhys | FSAM (Ours) | 1.30 ± 0.33 | 2.34 ± 0.60 | 1.75 ± 0.46 | **0.98 ± 0.04** | 7.83 ± 0.96 | 0.59 ± 0.02 |
| FactorizePhys (Ours) | FSAM (Ours) | **1.13 ± 0.36** | 2.42 ± 0.77 | **1.52 ± 0.50** | 0.97 ± 0.04 | 9.75 ± 1.05 | 0.65 ± 0.02 |
| Performance Evaluation on PURE Dataset, Subject-wise Split: Training (0.0 - 0.7), Test (0.7 - 1.0) | | | | | | | |
| PhysNet | - | 0.59 ± 0.27 | 1.28 ± 0.46 | 0.92 ± 0.44 | **1.00 ± 0.02** | **19.66 ± 1.18** | **0.90 ± 0.01** |
| PhysFormer | TD-MHSA* | 0.68 ± 0.26 | 1.31 ± 0.46 | 1.08 ± 0.43 | **1.00 ± 0.02** | 19.05 ± 1.07 | 0.87 ± 0.01 |
| EfficientPhys | SASN | **0.49 ± 0.26** | **1.21 ± 0.46** | **0.73 ± 0.42** | **1.00 ± 0.02** | 15.25 ± 1.20 | 0.80 ± 0.02 |
| EfficientPhys | FSAM (Ours) | 0.59 ± 0.27 | 1.28 ± 0.46 | 0.92 ± 0.44 | **1.00 ± 0.02** | 15.42 ± 1.25 | 0.80 ± 0.02 |
| FactorizePhys (Ours) | FSAM (Ours) | **0.49 ± 0.26** | **1.21 ± 0.46** | **0.73 ± 0.42** | **1.00 ± 0.02** | 19.63 ± 1.40 | 0.86 ± 0.01 |
| Performance Evaluation on UBFC-rPPG Dataset, Subject-wise Split: Training (0.0 - 0.7), Test (0.7 - 1.0) | | | | | | | |
| PhysNet | - | **1.62 ± 0.73** | **3.08 ± 1.16** | **1.46 ± 0.68** | **0.98 ± 0.06** | 5.21 ± 1.97 | 0.90 ± 0.01 |
| PhysFormer | TD-MHSA* | 1.76 ± 0.79 | 3.36 ± 1.30 | 1.60 ± 0.74 | 0.96 ± 0.08 | 6.10 ± 1.86 | 0.90 ± 0.01 |
| EfficientPhys | SASN | 2.30 ± 1.40 | 5.54 ± 2.53 | 2.28 ± 1.44 | 0.90 ± 0.13 | 6.75 ± 1.76 | 0.87 ± 0.01 |
| EfficientPhys | FSAM (Ours) | 2.91 ± 1.42 | 5.88 ± 2.52 | 2.79 ± 1.45 | 0.88 ± 0.14 | **6.79 ± 1.82** | 0.87 ± 0.01 |
| FactorizePhys (Ours) | FSAM (Ours) | 2.84 ± 1.42 | 5.87 ± 2.52 | 2.73 ± 1.46 | 0.88 ± 0.14 | 6.33 ± 2.00 | **0.91 ± 0.01** |

TD-MHSA*: Temporal Difference Multi-Head Self-Attention [77];

SASN: Self-Attention Shifted Network [37]; FSAM: Proposed Factorized Self-Attention Module

## A.6 Scalability Assessment of FSAM

We further investigate FSAM for its scalability to higher spatial-temporal resolution. For this, we perform within-dataset evaluation on the UBFC-rPPG dataset [2], which is pre-processed with the regular input dimension of $160 \times 72 \times 72$ as well as with a higher spatial and temporal dimension of $240 \times 128 \times 128$. Repeatable experiments are conducted with 10 different random seeds between 1 and 1000 to compare the performance of FactorizePhys with FSAM for each spatial-temporal input dimension.

Comparable performance, as observed in table 7, for both spatial-temporal input dimensions, suggests that FSAM can be easily deployed for different spatial-temporal scales. It should also be noted that the higher spatial dimension of video frames (i.e., $128 \times 128$) does not produce improved performance, indicating that the spatial dimension of $72 \times 72$ is sufficient to extract rPPG signals with end-to-end methods.

Table 7: Scalability Assessment of FSAM for Higher Spatial and Temporal Dimensions

| Input Dimension | Random Seed Value | MAE (HR) ↓ | | RMSE (HR) ↓ | | MAPE (HR)↓ | | Corr (HR) ↑ | | SNR ( dB, BVP) ↑ | | MACC (BVP) ↑ | |
|---|---|---|---|---|---|---|---|---|---|---|---|---|---|
| | | Mean | SE | Mean | SE | Mean | SE | Mean | SE | Mean | SE | Mean | SE |
| 160x72x72 | 10 | 2.84 | 1.43 | 5.87 | 2.52 | 2.73 | 1.45 | 0.88 | 0.14 | 6.49 | 2.03 | 0.90 | 0.01 |
| | 38 | 2.84 | 1.43 | 5.87 | 2.52 | 2.73 | 1.45 | 0.88 | 0.14 | 6.68 | 2.00 | 0.91 | 0.01 |
| | 55 | 2.97 | 1.41 | 5.89 | 2.52 | 2.84 | 1.44 | 0.88 | 0.14 | 6.52 | 1.98 | 0.91 | 0.01 |
| | 100 | 2.84 | 1.43 | 5.87 | 2.52 | 2.73 | 1.45 | 0.88 | 0.14 | 6.32 | 2.01 | 0.91 | 0.01 |
| | 128 | 2.97 | 1.41 | 5.89 | 2.52 | 2.84 | 1.44 | 0.88 | 0.14 | 6.42 | 1.99 | 0.91 | 0.01 |
| | 138 | 2.84 | 1.43 | 5.87 | 2.52 | 2.73 | 1.45 | 0.88 | 0.14 | 6.48 | 1.96 | 0.91 | 0.01 |
| | 212 | 2.91 | 1.42 | 5.88 | 2.52 | 2.79 | 1.45 | 0.88 | 0.14 | 6.40 | 1.99 | 0.91 | 0.01 |
| | 308 | 2.84 | 1.43 | 5.87 | 2.52 | 2.73 | 1.45 | 0.88 | 0.14 | 6.51 | 1.98 | 0.91 | 0.01 |
| | 319 | 2.91 | 1.42 | 5.88 | 2.52 | 2.79 | 1.45 | 0.88 | 0.14 | 6.44 | 2.03 | 0.91 | 0.01 |
| | 900 | 2.91 | 1.42 | 5.88 | 2.52 | 2.79 | 1.45 | 0.88 | 0.14 | 6.55 | 2.01 | 0.91 | 0.01 |
| | Mean | 2.89 | 1.42 | 5.88 | 2.52 | 2.77 | 1.45 | 0.88 | 0.14 | 6.48 | 2.00 | 0.91 | 0.01 |
| 240x128x128 | 10 | 3.04 | 1.92 | 7.56 | 3.64 | 3.22 | 2.18 | 0.83 | 0.17 | 6.68 | 1.93 | 0.90 | 0.01 |
| | 38 | 2.91 | 1.93 | 7.54 | 3.64 | 3.10 | 2.19 | 0.84 | 0.16 | 6.86 | 1.93 | 0.90 | 0.01 |
| | 55 | 2.97 | 1.92 | 7.54 | 3.64 | 3.16 | 2.18 | 0.84 | 0.16 | 6.63 | 1.91 | 0.91 | 0.01 |
| | 100 | 2.91 | 1.93 | 7.54 | 3.64 | 3.10 | 2.19 | 0.84 | 0.16 | 6.87 | 1.91 | 0.90 | 0.01 |
| | 128 | 3.11 | 1.91 | 7.56 | 3.64 | 3.28 | 2.17 | 0.84 | 0.17 | 6.71 | 1.87 | 0.90 | 0.01 |
| | 138 | 2.91 | 1.93 | 7.54 | 3.64 | 3.10 | 2.19 | 0.84 | 0.16 | 6.63 | 1.95 | 0.91 | 0.01 |
| | 212 | 3.04 | 1.92 | 7.56 | 3.64 | 3.22 | 2.18 | 0.83 | 0.17 | 6.81 | 1.93 | 0.90 | 0.01 |
| | 308 | 2.91 | 1.93 | 7.54 | 3.64 | 3.10 | 2.19 | 0.84 | 0.16 | 6.71 | 1.92 | 0.90 | 0.01 |
| | 319 | 3.04 | 1.92 | 7.56 | 3.64 | 3.22 | 2.18 | 0.83 | 0.17 | 6.83 | 1.93 | 0.91 | 0.01 |
| | 900 | 3.04 | 1.92 | 7.56 | 3.64 | 3.22 | 2.18 | 0.83 | 0.17 | 6.69 | 1.91 | 0.90 | 0.01 |
| | Mean | 2.99 | 1.92 | 7.55 | 3.64 | 3.17 | 2.18 | 0.84 | 0.17 | 6.74 | 1.92 | 0.90 | 0.01 |

## A.7 Multimodal rPPG Extraction

As iBVP dataset offers synchronized RGB and thermal infrared video frames, we conducted a brief experiment using FactorizePhys with FSAM to investigate whether combining both modalities can result in performance gains for the estimation of rPPG. For this, we also individually trained FactorizePhys on RGB and thermal frames keeping the identical data split of 70%-30%. Results

Table 8: Performance Evaluation on iBVP Dataset, Subject-wise Split: Train (70%), Test (30%)

| Modality of Input Frames | MAE (HR) ↓ | | RMSE (HR) ↓ | | MAPE (HR)↓ | | Corr (HR) ↑ | | SNR (dB, BVP) ↑ | | MACC (BVP) ↑ | |
|---|---|---|---|---|---|---|---|---|---|---|---|---|
| | Mean | SE | Mean | SE | Mean | SE | Mean | SE | Mean | SE | Mean | SE |
| T | 6.40 | 0.97 | 8.58 | 2.11 | 8.66 | 1.26 | 0.84 | 0.09 | -3.27 | 0.40 | 0.20 | 0.01 |
| RGB | 1.13 | 0.36 | 2.42 | 0.77 | 1.52 | 0.50 | 0.97 | 0.04 | 9.75 | 1.05 | 0.65 | 0.02 |
| RGBT | 1.10 | 0.36 | 2.42 | 0.77 | 1.49 | 0.50 | 0.97 | 0.04 | 9.65 | 1.04 | 0.64 | 0.02 |

in table 8 suggest weaker presence of rPPG signal in thermal infrared frames, leading to poorer performance when FactorizePhys is trained only on thermal frames, while not showing significant performance gains when jointly trained with RGB and thermal frames.

## A.8 Visual Overview of Cross-Dataset Generalization and Latency

Figure 5 offers a quick visual summary of the cross-dataset generalization performance on different evaluation metrics, their respective standard error, and latency for the proposed and existing SOTA

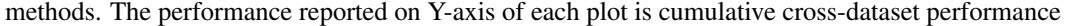

methods. The performance reported on Y-axis of each plot is cumulative cross-dataset performance

Figure 5: Cross-dataset performance comparison between SOTA and the proposed method reported with cumulative evaluation metrics, their standard error (SE) and latency

for respective models, averaged over different training and testing datasets. The proposed method outperforms existing state-of-the-art methods in all evaluation metrics by a significant margin, while achieving at-par latency.

## A.9   Computational Cost and Latency

Table 9 compares model parameters, latency on GPU and CPU, and model size of the proposed FactorizePhys with FSAM with that of the existing SOTA rPPG methods. Considering the identical inference time performance of the base FactorizePhys, when trained using the proposed FSAM, the proposed method uses an order of magnitude fewer parameters and achieves a par latency on both CPU and GPU systems. Relatively higher latency compared to the EfficientPhys [37] model, despite the fewer model parameters, is due to the difference in the number of floating point operations (FLOPS). FactorizePhys, being a 3D-CNN architecture, requires more FLOPS to compute 3D features at each layer compared to the fewer FLOPS for EfficientPhys [37] which implements the 2D-CNN

architecture. It should be noted that the FLOPS are also dependent on the input dimension, which is kept consistent for all the models. For resource critical deployment, FLOPS can be significantly reduced by decreasing the spatial dimension of input from $72 \times 72$ to $8 \times 8$ as found optimal for RTrPPG [3] or to $9 \times 9$ as used in the small branch of the Bigsmall model [44] for rPPG estimation.

Table 9: Comparison of FactorizePhys based on Model Parameters, Latency and Model Size

| Model | Model Parameters | Inference Time on CPU (ms)† | Inference Time on GPU (ms)‡ | Model Size (MB) |
|---|---|---|---|---|
| PhysFormer | 7380871 | 450.47 | 26.86 | 29.80 |
| PhysNet | 768583 | 272.89 | 1.36 | 3.10 |
| EfficientPhys with SASN | 2163081 | 371.08 | 1.31 | 8.70 |
| EfficientPhys with FSAM (ours) | 140655 | 82.19 | 5.62 | 0.57 |
| FactorizePhys Base (ours) | 51840 | 96.80 | 1.94 | 0.22 |
| FactorizePhys with FSAM (ours) | 52168 | 95.75 | 8.97 | 0.22 |

†CPU Specs: Intel® Core™ i7-10870H CPU @ 2.20GHz × 16 GB RAM.

‡GPU Specs: NVIDIA GeForce RTX 3070 Laptop GPU (CUDA cores = 5120).

## A.10  Visualization of Learned Attention

In fig. 6, we present additional samples of learned spatial-temporal features. For FactorizePhys trained with FSAM, we can observe superior cosine similarity and more relevant spatial distribution specifically under challenging scenarios with occlusions such as arising from hairs, eye-glasses and beard.

## A.11  Qualitative Comparison with Estimated rPPG Signals

Qualitative comparison of the estimated rPPG signals between the proposed method and the best performing SOTA method (i.e., EfficientPhys [37] is presented for different test datasets - iBVP [29] (fig. 7) , PURE [55] (fig. 8), and UBFC-rPPG [2] (fig. 9).

## A.12  Safeguards

We intend to release our rPPG estimation code only for academic purposes, with Responsible AI license (RAIL). Research areas that will benefit directly from this work include human-computer interaction and contactless health tracking or vital signs monitoring. Although the methods presented in this work may potentially benefit certain clinical scenarios, thorough validation studies, with appropriate ethics approval, are required to critically assess performance in such settings.

In addition, in some recent work, rPPG methods have been indicated as effective in detecting deep-fake videos. In this context, we would like to caution such a use, considering the main results presented for the models trained using the SCAMPS [42] dataset, consisting of synthesized avatars. We argue that the rPPG signal can be embedded in the synthesized (or deep-fake) videos, with a similar approach as used for generating the SCAMPS [42] dataset. In such scenarios, in spite of high accuracy in estimating rPPG signals, such methods can be fooled by the synthesized videos that embed BVP signals. Therefore, we highlight that it is necessary to use the rPPG signal estimation methods in this context with great caution.

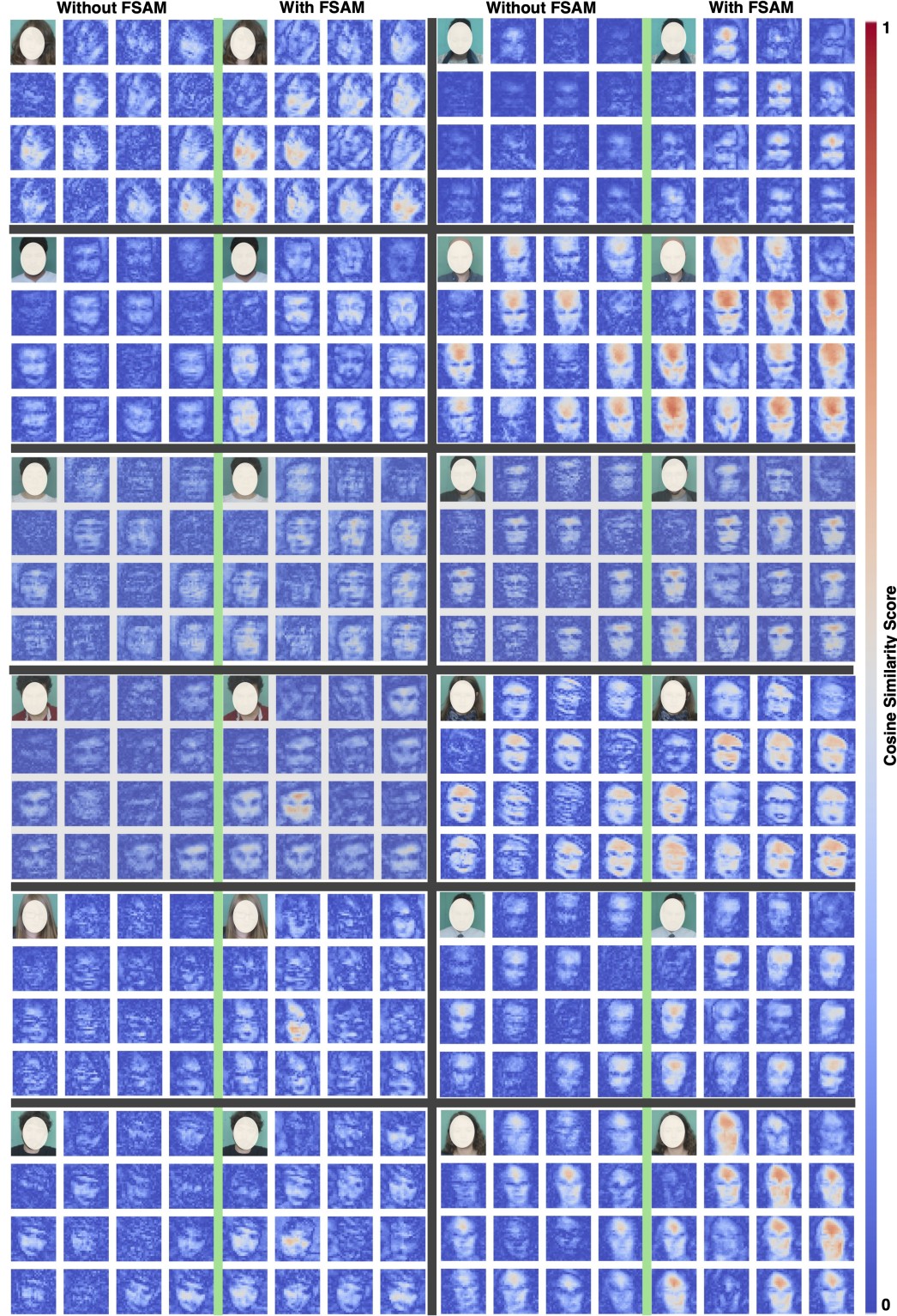

Figure 6: Visualization of Learned Spatial-Temporal Features

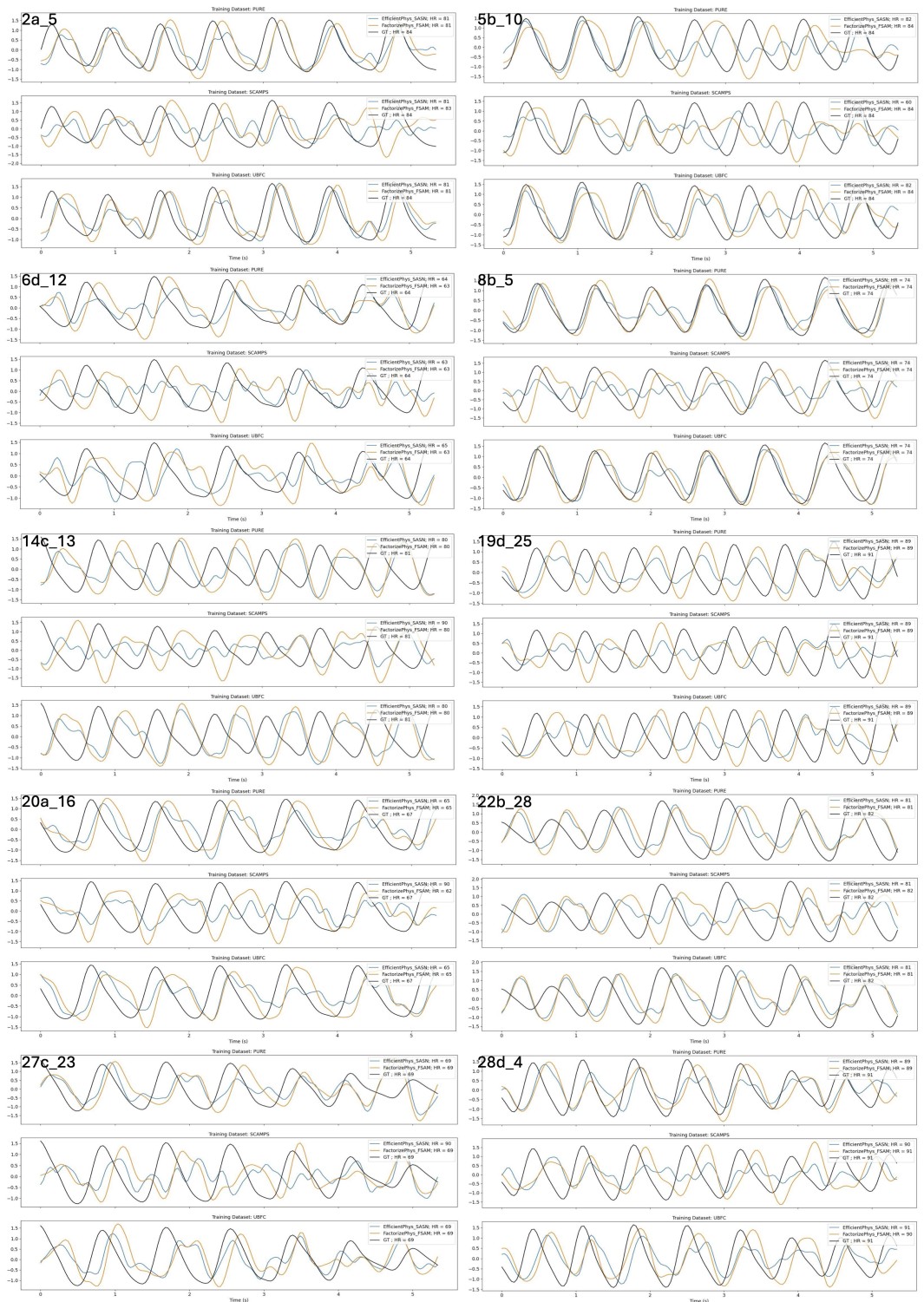

Figure 7: Comparison of Estimated rPPG Signals on iBVP Dataset for Models Trained with PURE, SCAMPS and UBFC-rPPG Datasets

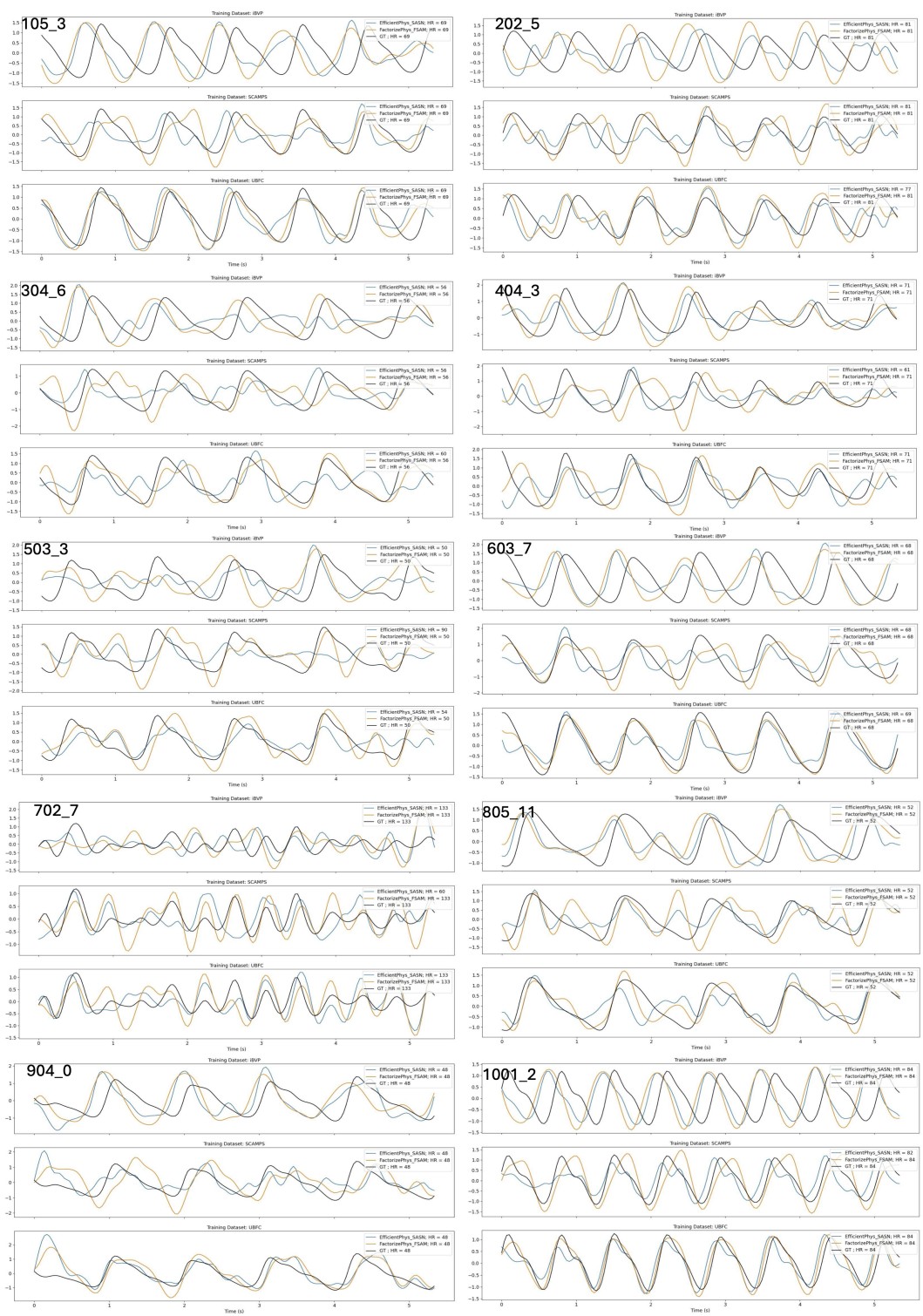

Figure 8: Comparison of Estimated rPPG Signals on PURE Dataset for Models Trained with iBVP, SCAMPS and UBFC-rPPG Datasets

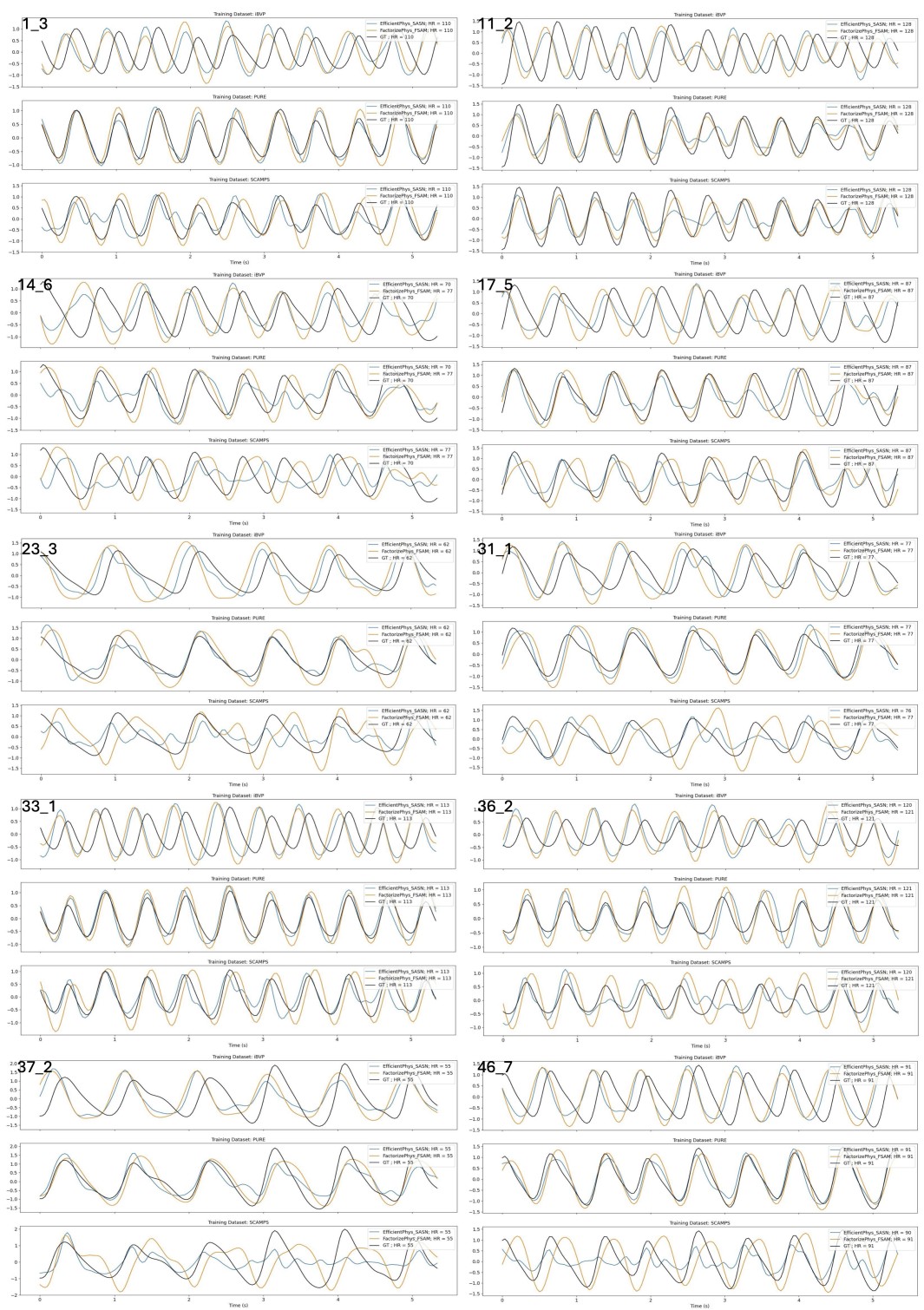

Figure 9: Comparison of Estimated rPPG Signals on UBFC-rPPG Dataset for Models Trained with iBVP, PURE and SCAMPS Datasets

