# OpenReview forum: "FactorizePhys: Matrix Factorization for Multidimensional Attention in Remote Physiological Sensing"
_NeurIPS.cc/2024/Conference — NeurIPS 2024 poster_

### Official Review · Reviewer_ECso · 2024-07-08

**Soundness:** 2
**Presentation:** 2
**Contribution:** 2
**Rating:** 3
**Confidence:** 4

**Summary:**

The paper introduces FactorizePhys that utilizes Non-negative Matrix Factorization (NMF) to decompose voxel embeddings. By integrating FSAM into both 3D-CNN and 2D-CNN architectures, FactorizePhys can estimate blood volume pulse signals from video frames. Through evaluations and comparisons with state-of-the-art rPPG methods, the effectiveness of FSAM and FactorizePhys are demonstrated.

**Strengths:**

1. Introducing the Factorized Self-Attention Module (FSAM) for computing multi-dimensional attention from voxel embeddings.
3. Evaluation of FSAM and FactorizedPhys against state-of-the-art rPPG methods
4. Integration of FSAM into existing 2D-CNN-based and 3D-CNN-based rPPG architectures to demonstrate its versatility.

**Weaknesses:**

1. The motivation to use the non-negative matrix factorization is unclear. Why can the factorized matrix be used as the attention? Authors should give more insights about the non-negative matrix factorization and show the attention maps to demonstrate the effectiveness.
2. There are other matrix factorization methods such as SVD and QR. Why is the non-negative matrix factorization chosen?
3. The factorization needs to solve an optimization problem which should do gradient descent steps. In line 216, the Authors only use a one-step gradient for matrix factorization. Is the one-step gradient enough to achieve satisfactory factorization?
4. The method part is not clearly illustrated. e.g., equation 5 is confusing, and the symbols in the equation are not well explained. The symbol mentioned in the main text is not well shown in the main figure.

**Questions:**

Please check the weakness part.

**Limitations:**

The authors mentioned the limitations of the work and proposed future directions such as time series estimation.

---

> ### Author Rebuttal · Authors · 2024-08-07
>
> We thank you for valuable comments and insightful questions. We will revise manuscript to reflect our responses.
>
> For your comments on visualization, we request you to refer our global response. We address rest of the comments here:
>
>
> * W1: The motivation to use the non-negative matrix factorization (NMF) as the attention:
>
> Factorization or matrix decomposition results in a low-rank approximation, and among such methods, NMF is highly researched area [1-4]. From prior works, Hamburger module [7] that implemented NMF, demonstrated its effectiveness in capturing global context, specifically for semantic segmentation and conditional image generation tasks. Hamburger module [7] outperformed transformer based network, though it was not implemented as attention mechanism, and the embeddings were limited to spatial and channel dimensions. Further, there have been efforts in reducing the redundancy in feature embeddings [6].
>
> These prior works inspire us to investigate if low-rank approximation of embeddings (intuitively - a type of squeeze operation [5],  that squeezes information without reducing dimension of the embeddings), when multiplied with the original embeddings (similar to excitation), forms an effective attention. The proposed FSAM module builds upon Hamburger module [7]  and extends NMF based factorization for multi-dimensional embeddings having spatial, temporal and channel dimensions. Optimization of a network having factorization module implemented as an attention mechanism, can influence the network to increase saliency of the relevant features, such that factorized approximation of embeddings retain these features, while discarding the less salient features. As factorization can handle multiple dimensions simultaneously with appropriate transformation or mapping, it underlines its unique potential as muti-dimensional attention mechanism, unlike existing candidates that reduces one or more dimensions of embeddings to compute attention.
>
> * W2: NMF v/s SVD, QR
>
> The rationale for choosing NMF compared to other decomposition techniques is as follows: i) The only constraint posed by NMF on the matrix, its vectors and the features is non-negativity, while SVD, QR and Vector Quantization (VQ) assume statistical independence or orthogonality between vectors of the approximated matrix. For deep-layer embeddings, constraints of orthogonality or statistical independence may not be relevant, and therefore such decomposition methods are not well-suited. ii) Owing to non-negativity or purely additive constraints, NMF effectively learns parts-based representation, which further enhances interpretability of the learned features [3].
>
> * W3: Gradient steps
>
> We fully agree with the reviewer that factorization needs to solve optimization problem. However, we would like to clarify in the manuscript that one-step gradient relates to a gradient step for each iteration that solves factorization with multiplicative update in one gradient step per iteration, as proposed in the Hamburger module [7].  Empirically we find 4 to 6 iterations to be sufficient to obtain the desired level of approximation. It is to be noted that our objective is not to achieve perfect fit with sub-zero approximation error. On the contrary, factorization serves as a better attention mechanism when it optimally approximates only salient features and discards less relevant features.
>
> * W4: Methods section, main figure and equation-5
>
> We will revise the methods section to sufficiently clarify the use of the symbols in the equations, along with clearer depiction of the symbols in the main figure. In regards to the equation 5, it mentions the negative Pearson correlation as an objective function for the downstream task of rPPG signal estimation task. We notice the typo in the description, and further understand the use of “i” and “T” in equation 5 can be clarified for better readability. To clarify here, “T” refers to total number of samples  of the estimated and ground-truth signal, which is also equal to the total number of input video frames.
>
>
>
> 1.	Y. -X. Wang and Y. -J. Zhang, "Nonnegative Matrix Factorization: A Comprehensive Review," doi: 10.1109/TKDE.2012.51
> 2.	Lee, D., & Seung, H. S. (2000). Algorithms for non-negative matrix factorization. Advances in neural information processing systems.
> 3.	Lee, Daniel D., and H. Sebastian Seung. (1999) "Learning the parts of objects by non-negative matrix factorization." nature 401.6755
> 4.	Gan, Jiangzhang, et al. "Non-negative matrix factorization: a survey." The Computer Journal 64.7 (2021): 1080-1092.
> 5.	Hu, Jie, Li Shen, and Gang Sun. "Squeeze-and-excitation networks." Proceedings of the IEEE conference on computer vision and pattern recognition, 2018
> 6.	Han, Zongyan, Zhenyong Fu, and Jian Yang. "Learning the redundancy-free features for generalized zero-shot object recognition." Proceedings of the IEEE/CVF conference on computer vision and pattern recognition. 2020.
> 7.	Z. Geng, M.-H. Guo, H. Chen, X. Li, K. Wei, and Z. Lin, ‘Is Attention Better Than Matrix Decomposition?’, in International Conference on Learning Representations, 2021.

---

> > ### Comment · Reviewer_ECso · 2024-08-12
> >
> > Thanks for the authors's response. My main concern still exists, which is how the NMF is related to the rPPG task. I think NMF is useful for general tasks, but the relation to rPPG is not that strong. Therefore, I will keep the score.

---

> > > ### Author Response · Authors · 2024-08-14
> > >
> > > We thank you for your comment and time to help improve our manuscript.
> > >
> > > While preparing for our previous response, we couldn’t spot out your main concern “which is how the NMF is related to the rPPG task”, which we are addressing below.
> > >
> > > To begin with, we fully agree that NMF is useful for general tasks or to put it in other words, it is agnostic to the downstream tasks. Here, we would like to highlight that rPPG estimation task is concerned with a series of spatial-temporal tasks including video object tracking, video segmentation and video action recognition from the perspective of learning spatial-temporal features. Therefore, rPPG estimation task benefits from the attention mechanism like any other spatial-temporal downstream task.
> > >
> > > As described in our response to Reviewer a4X8, the proposed approach helps jointly compute multi-dimensional attention using NMF to gain the modeling advantage, which is unlike most current methods that compute attention disjointly across different dimensions, which our reviewer highlighted as our strength.
> > >
> > > Further insights about the relevance of the proposed method for rPPG estimation task can be drawn from the visualization of learned attention maps, that we provided in Figure-4 of the PDF submitted along with global response. In these learned attention maps, higher cosine-similarity score can be observed for the model trained with the proposed FSAM module. Higher cosine-similarity score between the temporal dimension of the embeddings and the ground-truth PPG signal indicates higher saliency of temporal features. The spatial spread of high cosine-similarity scores highlights that the learned attention is selective to the regions of the face that have exposed skin surface (where rPPG signal can be found). This provides clearer evidence that the model trained with the FSAM module can appropriately pick the spatial features that are the sources of the desired temporal signal. Thus, the presented comparison of the visualization offers greater insights into the effectiveness of the joint computation of multi-dimensional attention for rPPG estimation task.
> > >
> > > On top of the joint computation of multi-dimensional attention using the proposed FSAM module, transformation of embeddings to factorization matrix as formalized in Equation-6 of the submitted manuscript plays a key role in obtaining significant performance gains as observed for our main, cross-dataset evaluation, which highlight superior generalization ability in the given task. Specifically, as per the Equation-6, temporal dimension of the embeddings is mapped to the vectors of factorization matrix, and the rest of the dimensions form features of the factorization matrix. This mapping enables explainable selection of rank of factorization in rPPG extraction case. Across the entire facial region, we expect only a single rPPG source signal, and similarly we expect this to be represented within the embeddings. Performing factorization of embeddings, with an optimally chosen rank (=1) is therefore highly suited for the given rPPG extraction task. Through an overall optimization of the network in presence of rank-1 approximation of embeddings, model learns to increase the saliency of the most relevant spatial-temporal features, which explains high effectiveness of the proposed method in rPPG estimation.
> > >
> > > We hope that the above explanation addresses the concern on “how the NMF is related to the rPPG task.”

---

### Official Review · Reviewer_c6jF · 2024-07-11

**Soundness:** 3
**Presentation:** 3
**Contribution:** 3
**Rating:** 6
**Confidence:** 3

**Summary:**

The paper presents a novel attention block FSAM devised for handling spatio-temporal data. It is benchmarked against a range of SOTA architectures on a suitable selection of different datasets, and found to perform strongly.

**Strengths:**

The paper is well presented and clearly structured

The is a good degree of novelty in the proposed architecture, and the application is of high interest.

The experiments are conducted over a well selected range of real world datasets.

A good level of detail is provided on the methodology, and the performance of the proposed approach appears strong.

**Weaknesses:**

While the paper's experimental results are promising, they do not appear to be accompanied by any uncertainty estimates. Including these is crucial to allow the reader to draw meaningful conclusions from the results.  I would refer the authors to question 6 in the checklist at the end of the paper. The answer given to the question 'does the paper report [...] statistical significance of the experiments' is given as "N/A", while the guidelines advise that this answer indicates that the paper does not include any experiments.

Also, while it is acceptable to focus on the intra-dataset performance within the main text, it is crucial to at least include the regular results in the Appendix.

A couple of minor typos:
On line 196:
"device a" -> "devise a"
On 236:
"It's" -> "Its"

**Questions:**

Is FSAM anticipated to have significantly wider impact on spatio-temporal applications, beyond rPPG?

How do the different candidate methods compare in their scalability to higher temporal and spatial resolutions?

**Limitations:**

There is a discussion on potential societal impact in the Appendix.

---

> ### Author Rebuttal · Authors · 2024-08-07
>
> We thank you for acknowledging the novelty of our contributions and significance of the reported results. We highly value your suggestions based upon which we will further revise manuscript.
>
> For your comments on statistical significance, uncertainty estimates and scalability, we request you to refer our global response. We address rest of the comments here:
>
>
> * Regular results in the Appendix:
>
> We would like to clarify that our main results are for cross-dataset evaluation, which offers better insights into the real-world performance for unseen data distribution. If we understand your suggestion correctly, it relates to reporting within-dataset performance. Based on this interpretation, we present intra-dataset/ within-dataset results in Table-7 in the PDF attached to the global response.
>
>
> * A couple of minor typos: On line 196: "device a" -> "devise a" On 236: "It's" -> "Its"
>
> Thank you for spotting the typos. We will address these and we will thoroughly proof-read the manuscript for its revision.
>
>
> * Is FSAM anticipated to have significantly wider impact on spatio-temporal applications, beyond rPPG?
>
> Among the existing works, NMF as implemented in the Hamburger module [1] demonstrated its effectiveness in capturing global context, specifically for the semantic segmentation and the conditional image generation tasks. While Hamburger module [1] outperformed transformer based network, authors did not implement their method as an attention mechanism, and the embeddings were limited to spatial and channel dimensions.
>
> FSAM builds upon it and investigates NMF based factorization of embeddings as multi-dimensional attention with additional temporal dimension.
>
> In principle, the factorization of embeddings results in a low-rank approximation of embeddings. Optimization of a network having factorization module implemented as an attention mechanism, influences the network to increase saliency of the most relevant features, such that factorized approximation of embeddings retain these features, while discarding the less salient features.
>
> As factorization can handle multiple dimensions simultaneously with appropriate transformation or mapping, it underlines its unique potential as muti-dimensional attention mechanism, unlike existing candidates.
>
> End-to-end estimation of rPPG signal from facial video frames represents one of the challenging spatial-temporal tasks as it requires network to learn to pick the spatial features having the desired temporal signature, while discarding the variance related to head-motion, illumination and skin-tones.
>
> While this work is limited to the evaluation of FSAM for a spatial-temporal task of estimating rPPG signal, based on the consistent performance gains across the datasets, and versatility of the module in 3D-CNN and 2D-CNN models, we envisage FSAM to have wider impact on spatio-temporal applications such as video segmentation, video object tracking, and action recognition, which we consider as future extension of this work.
>
>
> 1. Z. Geng, M.-H. Guo, H. Chen, X. Li, K. Wei, and Z. Lin, ‘Is Attention Better Than Matrix Decomposition?’, in International Conference on Learning Representations, 2021.

---

> ### Comment · Reviewer_c6jF · 2024-08-12
>
> Thank you for responding thoughtfully to my concerns. Overall my main concerns have been addressed so I shall update my score accordingly.
>
> > If we understand your suggestion correctly, it relates to reporting within-dataset performance.
>
> Yes that is correct, thank you for including these.
>
> Regarding the new Table 7:
>
> - Some of the STD values seem very large, for example the RMSE STD is around ten times larger than the mean. Is this a typo or just a feature of the dataset?
>
> - For the MAE values in "Performance evaluation on PURE", it seems as though Physnet should be bolded since it has 2.78, vs 2.83 which is currently in bold.

---

> > ### Author Response · Authors · 2024-08-14
> >
> > Thank you for finding our responses satisfactory and for increasing the score. Below we respond to your further queries:
> >
> > * Very large RMSE STD:
> >
> > Thanks for spotting out this error.
> >
> > We investigated this thoroughly and discovered an edge case scenario of one participant in the PURE dataset, with ground-truth HR=46. The data of only this participant was driving the RMSE STD very high. We first inspected the estimated rPPG signals for all the which were found well aligned with the ground-truth BVP signal.
> >
> > The root cause was low-cut-off freq (0.75 Hz), of the band-pass filter as implemented in rPPG-Toolbox [1], upon which our code is built. This filter is applied both to the ground-truth BVP signal and estimated rPPG signals before computing HR. This low-cut value impacts the FFT-peak based computation of HR for HR=46, as the main peak is suppressed and results are driven by the harmonics, leading to the observed large errors. We changed this to 0.5 Hz to accommodate low HR cases and re-evaluated ours as well as SOTA methods on PURE dataset, first for within-dataset case. We thoroughly verified that this change in low-cut-off frequency did not alter any outcome for HR higher than 46, which is the case with other datasets.
> >
> > Below, we present the revised within-dataset results for PURE dataset, where we observe expected RMSE STD range for all models.
> >
> > | Model                | Attention Module | MAE (HR) ↓ |  | RMSE (HR) ↓ |  | MAPE (HR)↓ |  | Corr (HR) ↑ |  | SNR ( dB, BVP) ↑ |  | MACC (BVP) ↑ |  |
> > | -------------------- | ------------------- | ---------- | ---------- | ----------- | ----------- | ---------- | ---------- | ----------- | ----------- | ---------------- | ---------------- | ------------ | ------------ |
> > |                      |                     | Mean       | STD        | Mean        | STD         | Mean       | STD        | Mean        | STD         | Mean             | STD              | Mean         | STD          |
> > | PhysNet              | \-                  | 0.88       | 0.50       | 2.29        | 4.17        | 1.25       | 0.63       | 0.99        | 0.03        | 22.25            | 1.99             | 0.92         | 0.01         |
> > | PhysFormer           | TD-MHSA\*           | 0.98       | 0.49       | 2.31        | 4.17        | 1.41       | 0.62       | 0.99        | 0.03        | 20.85            | 1.87             | 0.89         | 0.01         |
> > | EfficientPhys        | SASN                | 0.68       | 0.48       | 2.13        | 4.16        | 0.86       | 0.53       | 0.99        | 0.03        | 17.72            | 1.81             | 0.84         | 0.02         |
> > | EfficientPhys        | FSAM (Ours)         | 0.88       | 0.50       | 2.29        | 4.17        | 1.25       | 0.63       | 0.99        | 0.03        | 17.42            | 1.87             | 0.83         | 0.02         |
> > | FactorizePhys (Ours) | FSAM (Ours)         | 0.78       | 0.50       | 2.25        | 4.18        | 1.06       | 0.62       | 0.99        | 0.03        | 21.13            | 2.05             | 0.89         | 0.01         |
> >
> > To ensure the validity of our main results on PURE, we compared ours and the best-performing SOTA, trained using UBFC-rPPG.
> >
> > | Model                | Attention Module | MAE (HR) ↓ |  | RMSE (HR) ↓ |  | MAPE (HR)↓ |  | Corr (HR) ↑ |  | SNR ( dB, BVP) ↑ |  | MACC (BVP) ↑ |  |
> > | -------------------- | ---------------- | ---------- | ---------- | ----------- | ----------- | ---------- | ---------- | ----------- | ----------- | ---------------- | ---------------- | ------------ | ------------ |
> > |                      |                  | Mean       | STD        | Mean        | STD         | Mean       | STD        | Mean        | STD         | Mean             | STD              | Mean         | STD          |
> > | EfficientPhys        | SASN             | 3.39       | 1.58       | 12.59       | 103.66      | 3.65       | 1.35       | 0.84        | 0.07        | 10.27            | 1.17             | 0.69         | 0.02         |
> > | FactorizePhys (Ours) | FSAM (Ours)      | 0.54       | 0.21       | 1.70        | 1.56        | 0.77       | 0.31       | 1.00        | 0.01        | 15.18            | 0.96             | 0.80         | 0.02         |
> >
> > The large RMSE STD error of the SOTA method here is due to the low SNR of the estimated rPPG signal, affecting waveform morphology, and thereby the computed HR for the edge cases. As the SNR of rPPG signals estimated by FactorizePhys is higher, its results are further improved. This further underlines the superior cross-dataset generalization that our proposed model with FSAM module is able to achieve.
> >
> > Given these observations, we will further revise all cross-dataset results for evaluation performed on PURE dataset.
> >
> > * Bold for 2.78, vs 2.83:
> > Apologies for this typo. We will present the revised results with correctly “bolded” entries in the revised manuscript.
> >
> > [1] Liu, Xin, et al. "rppg-toolbox: Deep remote ppg toolbox." NeurIPS 2024

---

> > > ### Author Response · Authors · 2024-08-14
> > >
> > > In continuation of the previous comment, below we share revised main results (after changing the low-cutoff for band-pass filtering) for all the models trained using UBFC-rPPG dataset and evaluated on PURE dataset.
> > >
> > >
> > > | Model                | Attention Module | MAE  |  (HR) ↓ | RMSE  | (HR) ↓ | MAPE  | (HR)↓ | Corr | (HR) ↑ | SNR   | (dB, BVP) ↑ | MACC | (BVP) ↑ |
> > > | -------------------- | ---------------- | ---- | ------- | ----- | ------ | ----- | ----- | ---- | ------ | ----- | ----------- | ---- | ------- |
> > > |                      |                  | Mean | STD     | Mean  | STD    | Mean  | STD   | Mean | STD    | Mean  | STD         | Mean | STD     |
> > > | PhysNet              | \-               | 7.70 | 2.16    | 18.30 | 113.63 | 13.15 | 3.87  | 0.66 | 0.10   | 11.48 | 1.12        | 0.74 | 0.02    |
> > > | PhysFormer           | TD-MHSA\*        | 7.47 | 2.18    | 18.33 | 131.60 | 11.72 | 3.47  | 0.65 | 0.10   | 9.22  | 1.12        | 0.68 | 0.02    |
> > > | EfficientPhys        | SASN             | 3.39 | 1.58    | 12.59 | 103.66 | 3.65  | 1.35  | 0.84 | 0.07   | 10.27 | 1.17        | 0.69 | 0.02    |
> > > | EfficientPhys        | FSAM (Ours)      | 2.10 | 1.19    | 9.37  | 76.23  | 2.60  | 1.18  | 0.92 | 0.05   | 11.25 | 1.07        | 0.71 | 0.02    |
> > > | FactorizePhys (Ours) | FSAM (Ours)      | 0.54 | 0.21    | 1.70  | 1.56   | 0.77  | 0.31  | 1.00 | 0.01   | 15.18 | 0.96        | 0.80 | 0.02    |
> > >
> > > \* TD-MHSA*: Temporal Difference Multi-Head Self-Attention \cite{yu2022PhysFormer};  SASN: Self-Attention Shifted Network \cite{liu2023EfficientPhys}; FSAM: Proposed Factorized Self-Attention Module
> > >
> > >
> > > We further notice the large RMSE STD errors for SOTA methods attributed to the low SNR of the estimated rPPG signal, affecting waveform morphology, and thereby the computed HR for the edge cases. As the SNR of rPPG signals estimated by FactorizePhys is higher, it significantly outperforrms on edge cases and unseen datasets. The superior cross-dataset generalization of our proposed FSAM module is further evidenced with these revised results.

---

### Official Review · Reviewer_a4X8 · 2024-07-22

**Soundness:** 3
**Presentation:** 3
**Contribution:** 2
**Rating:** 6
**Confidence:** 4

**Summary:**

The paper proposed the Factorized Self-Attention Module (FSAM), which jointly computes multi-dimensional attention across spatial, temporal, and channel dimensions using non-negative matrix factorization (NMF). The FSAM is integrated into a new end-to-end 3D-CNN architecture called FactorizedPhys, designed to estimate blood volume pulse signals from video frames. Extensive experiments on multiple datasets demonstrate FSAM's effectiveness.

**Strengths:**

The overall writing of the paper is clear. The proposed method leverages the strengths of matrix factorization to capture global spatial-temporal context effectively. Specifically:

- Unlike most current methods that compute attention disjointly across different dimensions, the proposed method jointly computes multi-dimensional attention using non-negative matrix factorization to gain the modeling advantage.

-Ablation studies are conducted to assess the different mappings of voxel embeddings in the factorization matrix.

- Model Complexity is analyzed experimentally.

**Weaknesses:**

-Lack of visualization to show how the results of the proposed method are superior (say, an image of the output to show the learned attention).

-Lack of variance of the results in the experiments table.

-It is still unclear why and in what situation the proposed method is better than the SOTA methods. Say, pick some cases from the model output that show that FactorizePhys is better than the others.

**Questions:**

- Are the methods sensitive to the hyperparameters? Lack of sensitivity analysis.

-The authors could consider visualizing more about the outputs of the models.

-Lack of scalability analysis of the methods, especially when the NMF is adopted. For large datasets, it is valuable to know whether the method is scalable compared to other methods.

**Limitations:**

The limitations are mentioned in the paper but could be highlighted more clearly in the discussion section.

---

> ### Author Rebuttal · Authors · 2024-08-07
>
> We thank you for acknowledging the strengths of our contributions, providing constructive feedback and suggestions. We will revise manuscript to reflect responses to your comments that we describe here.
>
> For your comments on visualization, variance and scalability, we request you to refer our global response. We address rest of the comments here:
>
> * Describing the cases when the proposed method is better than the SOTA methods:
> We will revise the discussion section to describe the key findings as follows:
>
> For all reported evaluation metrics, the proposed method outperforms SOTA methods for its cross-dataset evaluation (revised Table-1 in PDF) on PURE and iBVP datasets, for all respective training datasets. This suggest consistent and superior generalization achieved by the proposed method. Cross-dataset evaluation on UBFC-rPPG dataset further highlights the performance gains of the proposed FactorizePhys model when it is trained using iBVP and SCAMPS datasets, and at-par performance with SOTA models when trained using PURE dataset. FactorizePhys uniquely outperforms the SOTA methods on all testing datasets when SCAMPS dataset is used for training, further stressing the superior generalization achieved from synthesized dataset with the proposed method. Lastly, the proposed method consistently offers superior SNR and MACC for the estimated rPPG signals, which highlights enhanced reliability. We further report intra-dataset evaluation (Table-7 in PDF), where performance of FactorizePhys is at-par with the SOTA methods.
>
> * Are the methods sensitive to the hyperparameters? Lack of sensitivity analysis
>
> While we follow fair training-testing strategy as described below and report statistical significance of results in the supplementary data, we specifically evaluated sensitivity for factorization rank – which is the most important hyper-parameter for the proposed FSAM module. The results shown in the appendix Table 4 of the submitted manuscript indicate low sensitivity for a set of low-ranks, while the performance drops with higher rank, which is an expected behaviour. High-rank factorization results in low matrix approximation error which does not remove redundancy in the embeddings, and thereby not serving as an effective attention mechanism. Factorization rank is therefore an important hyperparameter that is required to be adjusted based on the architecture, placement of the module within the architecture and the downstream task. For rPPG signal extraction downstream task, we expect only one underlying signal source and therefore rank-1 factorization of temporal vectors offers optimal performance.
>
> Fair training-testing strategy: All model-specific hyper-parameters were maintained as provided by the respective SOTA methods, while the training pipeline related hyper-parameters were kept consistent for training all the models. Training pipe-line related hyper-parameters that we maintained consistent include - pre-processing steps for images and labels, batch-size, number of epochs, learning rate, scheduler and optimizer. However, we noticed that the number of epochs that we initially kept as 30, resulted in extremely low training-loss for all SOTA methods, affecting their cross-dataset generalization. The proposed method did not show such an overfitting problem and was not found senstive to the number of training epochs after convergence. This offered advantage to the proposed method. For fair evaluation against SOTA methods, we revised the training, validation and testing strategy, in which instead of using the validation set based best epoch from 30 epochs training, we re-trained all models for 10 epochs, similar to a recent work [1], and used the last epoch for cross-dataset as well as within-dataset evaluation. Table 6 in PDF highlights that the updated strategy offers fair evaluation against the SOTA methods. Revised Table 1 and new appendix Table 7 provide detailed results for cross-dataset and regular (within-dataset) evaluation respectively, with statistical variance mentioned for all the evaluation metrics.
>
>
> * The authors could consider visualizing more about the outputs of the models.
>
> We appreciate this feedback and we will add a figure in the revised manuscript to show the learned attention map as well as output of the models. Here, in Figure 5 of the PDF, we have added a sample plot that compares the output rPPG signals of the proposed FactorizePhys (in orange) and EfficientPhys (best performing SOTA model, in blue) with the ground-truth BVP signal (GT, in black). We have provided large sample set of such comparisons in the supplementary data shared with the ACs. To fit more plots in the supplementary data, we used higher JPEG compression for the figures, due to which images may appear noisy, for which we apologize.
>
>
> * The limitations are mentioned in the paper but could be highlighted more clearly in the discussion section.
>
> We appreciate reviewer for this comment. Following are few limitations which we would like to clearly highlight in the discussion section:
>
> (a) While the proposed FSAM module has shown to be effective spatial-temporal attention for rPPG signal extraction task, its efficacy as an attention mechanism for other spatial-temporal tasks such as video understating, video object tracking, and video segmentation, needs to be further investigated.
>
> (b) For signal extraction tasks, different forms of constrained NMF were not investigated. Specifically the variants of NMF which put temporal or frequency constraints on the time-series vectors may offer more effective attention, since such constraints can be chosen based on the characteristics of the ground-truth signal to be estimated.
>
>
>
>
> 1. C. Zhao, H. Wang, H. Chen, W. Shi and Y. Feng, "JAMSNet: A Remote Pulse Extraction Network Based on Joint Attention and Multi-Scale Fusion," 2023, doi: 10.1109/TCSVT.2022.3227348.

---

> > ### Comment · Reviewer_a4X8 · 2024-08-12
> >
> > The authors answer most of my questions. I especially appreciate the effort of putting in more visualization.  But the concern "It is still unclear why and in what situation the proposed method is better than the SOTA methods" is still not answered well. I understand the experimental results are better than the SOTA but still not clear how the proposed method could gain significant improvement. If it is due to the use of joint computation of multi-dimensional attention, the case study should compare the difference(either visualization or some deep explanation of results) of different approaches that use attention as well.

---

> > > ### Author Response · Authors · 2024-08-14
> > >
> > > We sincerely thank you for appreciating our efforts on preparing visualization of learned attention maps. We also apologize for not making it sufficiently clear about why the proposed FSAM module achieves significant performance gains. Here we would like to provide further insights.
> > >
> > > Firstly, aligned with your suggestion, through our visualization presented in Figure-4 of the PDF submitted with the global response, we compare embeddings of the identical base 3D CNN model – trained without and with FSAM module (which performs joint computation of multi-dimensional attention) .
> > >
> > > In these visualized learned attention maps, higher cosine-similarity score can be observed for the model trained with the proposed FSAM module. Higher cosine-similarity score between the temporal dimension of the embeddings and the ground-truth PPG signal indicates higher saliency of temporal features. Further look at the spatial spread of high cosine-similarity scores highlights that the learned attention is selective to the regions of the face with the exposed skin surface (where rPPG signal can be found). This provides a clearer evidence that the model trained with the FSAM module can appropriately pick the spatial features that are the sources of the desired temporal signal. Thus, the presented comparison of the visualization offers greater insights into the effectiveness of the joint computation of multi-dimensional attention.
> > >
> > > To compare the performance against existing attention modules, we picked the best performing SOTA (EfficientPhys that implements SASN attention) based on our main results and adapted our proposed FSAM for 2D-CNN based EfficientPhys architecture. We then replaced SASN module with the proposed FSAM module. Our main (cross-dataset) results, in Table-1 of PDF and within-dataset results (Table-7) of the PDF in global response, provide direct comparison of the performance of FSAM and SASN in EfficientPhys architecture. While EfficientPhys with FSAM module performs at-par with the EfficientPhys having SASN attention module, it can be inferred that 2D-CNN based architecture is not able to leverage the full potential of the joint spatial-temporal attention, which is achieved in the proposed 3D-CNN based architecture.
> > >
> > > We understand that in addition to the joint computation of multi-dimensional attention using the proposed FSAM module, transformation of embeddings to factorization matrix as formalized in Equation-6 of the submitted manuscript plays a key role in obtaining significant performance gains as observed for our main, cross-dataset evaluation, which highlight superior generalization ability. Specifically, as per the Equation-6, temporal dimension of the embeddings is mapped to the vectors of factorization matrix, and the rest of the dimensions form features of the factorization matrix. This mapping enables explainable selection of rank of factorization in rPPG extraction case. Across the entire facial region, we expect only a single rPPG source signal, and similarly we expect this to be represented within the embeddings. Performing factorization of embeddings, with an optimally choosen rank (=1) is therefore highly suited for the given rPPG extraction task. Through an overall optimization of the network in presence of rank-1 approximation of embeddings, model learns to increase the saliency of the most relevant features, which explains high effectiveness of the proposed method.
> > >
> > > We will revise the results and discussion section to offer these deeper insights.
> > >
> > > Hope above response address your main concern.

---

### Author Rebuttal · Authors · 2024-08-07

We would like to sincerely thank all reviewers for their valuable feedback that helps stregthening our contributions. We would like to respond to common comments here, while responding individually for the rest.

* a4X8, ECso:
Visualization of the attention maps to demonstrate the effectiveness the proposed FSAM module:

We fully agree that visualization of learned attention can provide insight into the efficacy of the proposed method. In figure 4 of PDF, we have added the same for the network trained without and with the proposed FSAM module. Each tile represents a channel of 4D embedding (spatial, temporal and channel), and the figure shows all the channels of an embedding layer. For each channel, we compute cosine similarity between the temporal dimension of a spatial-temporal voxel and the ground-truth signal. The resultant color-coded cosine-similarity score-map offers more intuitive visualization of the learned spatial-temporal attention, as compared to that used in the existing works [1] that disjointly present spatial and temporal attention. Given this comment, we will introduce a new figure in the results section of revised manuscript, along with a brief description on our approach to generate visualization of the learned attention map. Further, our supplementary data with more samples of learned embeddings, model outputs of estimated signals, detailed results and code is shared with the ACs.

* a4X8, c6jF:
Reporting of statistical significance of the experiments, variance and uncertainty estimates:

To report statistical significance, we picked the best performing SOTA method (EfficientPhys [2]) from the main cross-dataset evaluation. For the SOTA and the proposed method, we performed 10 rounds of training and testing with random seed values. Paired T tests show that the proposed method outperforms SOTA with statistical significance obtained for all reported evaluation metrics. Results are added to an excel-sheet that is a part of the supplementary data, which we have recently shared with the ACs.

For rPPG signal estimation task, total uncertainty measure has been shown to highly correlate with the absolute error obtained from the heart-rate computed using the estimated rPPG signal and the ground-truth heart-rate [3]. While authors used CHROM [5] rPPG method to show this correlation, they clarified that their method is agnostic to the rPPG method [3] and their findings can be generalized to time-series estimation task [3, 4]. Based on this understanding, we report mean absolute error as well as mean values of other relevant rPPG specific evaluation metrics in the submitted manuscript. We have further revised Table-1 (in attached PDF) to provide the corresponding variance measures to draw meaningful conclusions from the experiments. We understand that the reported evaluation metrics along with the variance measures thoroughly compare the uncertainty estimates of the proposed method with that of the SOTA methods.

* a4X8, c6jF:
Scalability Analysis:

i) Performance of FSAM for higher temporal and spatial resolution:
In new appendix Table-8 (in PDF), we compare the performance of the proposed module by varying spatial resolution and temporal dimension. With higher spatial resolution and more temporal frames, we observe small performance gain. We further conducted repeated tests (10 paired tests with different seed values) that showed non-significant performance difference between the models trained with 160x72x72x3 (low-res) and 240x128x128x3 (high-res) dimensional input video frames. Results of the repeated tests for scalability are also added to the supplementary data.

ii) Overview of computational complexity when FSAM is adapted for higher temporal and spatial resolution:
The number of trainable parameters of the FSAM module are dependent only on the parameters in pre and post convolution layers, while the factorization operation is implemented as no-grad, thereby not adding trainable parameters as well as not directly affecting the gradients flow for optimization of the main network.
For low-res and high-res inputs, trainable parameters of FSAM are just 328 within our implemented 3D-CNN architecture having 56200 parameters. For low-res and high-res, dimension of embeddings approximated by FSAM are 160x392 and 240x5000 respectively, which are both approximated in 4 iterations, executed only during forward pass. For very high-resolution embeddings, one of the high-resolution dimensions can be appropriately splitted to execute batched optimization for factorization. Thus FSAM adds negligible overhead making it highly suitable for scalability.
Our additional ablation study (in supplementary data) highlight that the models trained using the FSAM retain the same performance, when deployed without the FSAM module for evaluation. This eliminates inference time latency and computational overhead of the module. The results reported in the revised Table-1 (in PDF) are obtained from the proposed FactorizePhys model deployed without FSAM module. This further improves the scalability of the proposed method.


1. C. Zhao, H. Wang, H. Chen, W. Shi and Y. Feng, "JAMSNet: A Remote Pulse Extraction Network Based on Joint Attention and Multi-Scale Fusion," June 2023, doi: 10.1109/TCSVT.2022.3227348
2. Liu, Xin, et al. "Efficientphys: Enabling simple, fast and accurate camera-based cardiac measurement." Proceedings of the IEEE/CVF winter conference on applications of computer vision. 2023
3. R. Song, H. Wang, H. Xia, J. Cheng, C. Li and X. Chen, "Uncertainty Quantification for Deep Learning-Based Remote Photoplethysmography," 2023, doi: 10.1109/TIM.2023.3317379.
4. W. Qian, D. Zhang, Y. Zhao, K. Zheng and J. J. Q. Yu, "Uncertainty Quantification for Traffic Forecasting: A Unified Approach," 2023, doi: 10.1109/ICDE55515.2023.00081.
5. De Haan, Gerard, and Vincent Jeanne. "Robust pulse rate from chrominance-based rPPG." IEEE transactions on biomedical engineering 60.10 (2013): 2878-2886.

---

### Comment · Area_Chair_kNQ6 · 2024-08-13
**Source code from authors**

Please find the code from authors

https://drive.google.com/file/d/1agSB6Ez4ywA7E-rH3Bn2OOdp4WbVaU3h/view?usp=drive_link

---

### Decision · Program_Chairs · 2024-09-25

**Decision:**

Accept (poster)

**Comment:**

The paper is well-written and clearly presents a novel method that leverages the strengths of Non-negative Matrix Factorization (NMF) to effectively capture global spatial-temporal context. While NMF is powerful for vision tasks, the authors apply it within a new end-to-end 3D-CNN architecture aimed at estimating blood volume pulse signals from video frames.

The reviewers unanimously agree that the empirical results on remote Photoplethysmography (rPPG) are promising.

However, to further strengthen the paper, the authors are encouraged to clarify the motivation behind using NMF in this context. Incorporating additional details such as the visualization of attention maps and providing statistical significance, as suggested by the reviewers, would enhance the paper's robustness and the clarity of its contributions.

I recommend accepting this paper, provided the authors address these points in their final submission.